# Small RNA pathways in the nematode *Ascaris* in the absence of piRNAs

Maxim V. Zagoskin ⬤ [1,2,3,5], Jianbin Wang ⬤ [1,2,3,4,5 ✉], Ashley T. Neff[1], Giovana M. B. Veronezi[1] & Richard E. Davis ⬤ [1,2 ✉]

Small RNA pathways play key and diverse regulatory roles in *C. elegans*, but our understanding of their conservation and contributions in other nematodes is limited. We analyzed small RNA pathways in the divergent parasitic nematode *Ascaris*. *Ascaris* has ten Argonautes with five worm-specific Argonautes (WAGOs) that associate with secondary 5'-triphosphate 22-24G-RNAs. These small RNAs target repetitive sequences or mature mRNAs and are similar to the *C. elegans* mutator, nuclear, and CSR-1 small RNA pathways. Even in the absence of a piRNA pathway, *Ascaris* CSR-1 may still function to "license" as well as fine-tune or repress gene expression. *Ascaris* ALG-4 and its associated 26G-RNAs target and likely repress specific mRNAs during testis meiosis. *Ascaris* WAGO small RNAs demonstrate target plasticity changing their targets between repeats and mRNAs during development. We provide a unique and comprehensive view of mRNA and small RNA expression throughout spermatogenesis. Overall, our study illustrates the conservation, divergence, dynamics, and flexibility of small RNA pathways in nematodes.

---

[1] Department of Biochemistry and Molecular Genetics, University of Colorado School of Medicine, Aurora, CO, USA. [2] RNA Bioscience Initiative, University of Colorado School of Medicine, Aurora, CO, USA. [3] Department of Biochemistry and Cellular and Molecular Biology, University of Tennessee, Knoxville, TN, USA. [4] UT-ORNL Graduate School of Genome Science and Technology, University of Tennessee, Knoxville, TN, USA. [5]These authors contributed equally: Maxim V. Zagoskin, Jianbin Wang. ✉email: jianbin.wang@utk.edu; richard.davis@cuanschutz.edu

   

Small RNAs contribute to many regulatory processes. They are associated with diverse functions including repressing foreign invaders and mobile elements, transcriptional regulation, mRNA translation and degradation, DNA repair, chromatin regulation and epigenetic inheritance, and ciliate genome rearrangements[1–5]. One of the first discoveries of small RNAs and their role in gene regulation was in the free-living nematode *Caenorhabditis elegans*[6,7]. Subsequent studies in *C. elegans* have revealed a diverse and complex set of small RNAs, pathways, and associated Argonautes[8–11].

*C. elegans* small RNAs include miRNAs, 21U-RNAs (piRNAs) and small-interfering RNAs (siRNAs). *C. elegans* miRNAs are involved in post-transcriptional gene regulation through regulation of mRNA translation and degradation[12] but have also been associated with transcriptional activation[13]. *C. elegans* can generate siRNAs against foreign elements and also has a large repertoire of endogenous siRNAs[8–11]. These *C. elegans* small RNAs are named based on their length and the predominant first nucleotide of the RNA, e.g., 21U-, 22G-, and 26G-RNAs. 21U-RNAs have a 5'-monophosphate and 3' 2'-O-methylation[14–16]. They are primarily thought to identify foreign (non-self) RNA elements. The identification of these foreign elements through RNA base-pairing leads to the subsequent synthesis of secondary siRNAs that repress their targets. 22G-RNAs have a 5'-triphosphate[16–20]. These are secondary siRNAs as they are generated in response to other small RNAs (21U-RNA, 26G-RNA, or other siRNAs)[20,21]. Thus, primary siRNAs base-pair with transcript targets, marking or identifying them for the synthesis of the more abundant antisense, secondary 22G-RNAs by RNA-dependent RNA polymerases (RdRPs)[18–23]. 22G-RNAs silence mobile elements, pseudogenes, non-annotated loci, and select endogenous, germline genes[18]. They also serve to "activate or license", "tune", or repress gene expression[24–27]. 26G-RNAs have a 5' monophosphate[16,19,28,29]. There are two classes of 26G-RNAs in *C. elegans*, one is testis-specific and associated with the Argonautes ALG-3/4[30–32], the other is expressed in early embryos and associated with the Argonaute ERGO-1[28,29,32,33]. Like 21U-RNAs, 26G-RNAs trigger or act to prime the synthesis of secondary siRNAs (22G-RNAs) through base-pairing with targets[8–11]. Overall, *C. elegans* secondary 22G-RNAs are amplified responses to targets identified by 21U- and 26G-RNAs and other primary siRNAs.

Small RNAs are bound by effector Argonaute proteins. *C. elegans* Argonautes have undergone significant expansion and diversification. Twenty-seven Argonaute genes were originally described[23]; 19 are expressed (Julie Claycomb, personal communication). These include members of the AGO, PIWI, and WAGO (Worm-specific Argonautes) Argonaute clades[34]. The *C. elegans* AGO-clade Argonautes (5 AGOs) interact with miRNAs and 26G-RNAs, the PIWI clade (1 AGO) with 21U-RNAs (the worm piRNA ortholog), and the WAGO-clade (13 AGOs) with 22G-RNAs. Several of the WAGO clade Argonautes function in the nucleus regulating transcription and heterochromatin formation[3,11,35,36]. The largest expansion of *C. elegans* Argonautes is in the WAGO clade[23]. Many of these WAGOs are thought to have redundant functions in *C. elegans*.

Nematodes are an extremely diverse and abundant phylum adapted to a wide variety of lifestyles[37–39]. While extensive analyses of *C. elegans* Argonautes, small RNAs, and pathways have been carried out, relatively little is known regarding the conservation, divergence, or function of these pathways in other nematodes[8,40,41]. Nematodes have been divided into three major classes and five clades with *C. elegans* and its relatives as members of Clade V[42–45]. The nematode *Ascaris* is a parasite of humans (and pigs) infecting upwards of 800,000 people[46–48]. *Ascaris* is a Clade III nematode estimated to have diverged from *C. elegans*

~365–400 million years ago[49,50]. Previous studies in *Ascaris* indicated that piRNAs and PIWI Argonautes are absent in *Ascaris*[51]. PIWI Argonautes and piRNAs play a key role in repressing mobile elements in the germline. *C. elegans* piRNAs have been proposed to serve in identifying and defining foreign genetic elements acting upstream of secondary siRNA pathways[52–56]. Thus, they have been described as defining self vs. non-self[57]. The WAGO-associated 22 G secondary RNAs function in silencing and can maintain silencing of the foreign elements over generations in *C. elegans*[8,11]. To counteract silencing, it has been proposed that 22G-RNAs associated with the *C. elegans* CSR-1 function to 'license', identify self, or protect germline genes from repression and allow for their expression[24,26,27]. Given the key role of piRNAs in many organisms, and their role in *C. elegans*, the absence of piRNAs and PIWI in *Ascaris* raises the question of how *Ascaris* small RNA pathways have adapted to the absence of piRNAs and PIWI (e.g., self vs non-self). Does *Ascaris* still need small RNA pathways to "license" gene expression without the presence of piRNAs? How is self vs non-self elements determined?

Here, we generated antibodies to *Ascaris* Argonaute proteins and carried out Argonaute IP and small RNA sequencing to characterize small RNA pathways. Two *Ascaris* Argonautes are highly specific for binding miRNAs (AsALG-1) and 26G-RNAs (AsALG-4). AsWAGO Argonautes bind 5'-triphosphate small RNAs (22–24G-RNAs). These small RNAs target repetitive sequences including mobile elements, but they also target mRNAs and likely "license", "tune", and/or repress gene expression. Two *Ascaris* WAGOs change their genomic targets (mRNAs vs repetitive sequences) in different developmental stages. We exploited the long ~1 m *Ascaris* male germline to obtain discrete regions of the testis and analyzed *Ascaris* Argonautes and their small RNAs throughout spermatogenesis. These analyses provide a unique and comprehensive timeline for the expression of Argonautes, their bound small RNAs and targets, and changes in expression of their corresponding genomic or mRNA targets throughout nematode spermatogenesis. Our study suggests that in the absence of piRNAs and with extensive evolutionary divergence from *C. elegans*, several *Ascaris* small RNA pathways and Argonautes appear to bind similar small RNAs, have similar targets, and in many cases appear to serve similar roles in both nematodes. Therefore, the potential "licensing" or fine-tuning of gene expression by *Ascaris* CSR-1 does not appear to be a consequence or counter-response to the presence of piRNAs. Overall, our studies illustrate the conservation and divergence of small RNA pathways that illustrate the flexibility and adaptability of Argonautes and small RNA pathways in nematodes.

## Results

***Ascaris* Argonautes**. We previously identified 10 *Ascaris* Argonautes[51]. Alignment and phylogenetic analyses of these Argonautes indicates the presence of 5 AGO-clade and 5 WAGO (Worm-specific Argonautes) Clade Argonautes, but the absence of PIWI Argonautes (Fig. 1A). Of the 5 *Ascaris* AGOs, two (AsALG-1 and AsALG-6) cluster with *C. elegans* miRNA associated Argonautes and one (AsALG-4) with *C. elegans* ALG-3/4 Argonautes. The two additional Argonautes (AsALG-5 and AsALG-7) cluster with 26G-RNA-like Argonautes or RDE-1 (Note: AsALG-5 is not orthologous to *C. elegans* AsALG-5) (Fig. 1). These *Ascaris* AGOs have conserved RNase H, catalytic tetrad residues that confer slicer activity except for AsALG-6.

The 5 *Ascaris* WAGOs were named AsCSR-1, AsWAGO-1, AsWAGO-2, AsWAGO-3, and AsNRDE-3 based on expression pattern, their sequence and phylogenetic similarity to *C. elegans*, and the small RNAs associated with these Argonautes (see below) (Fig. 1).

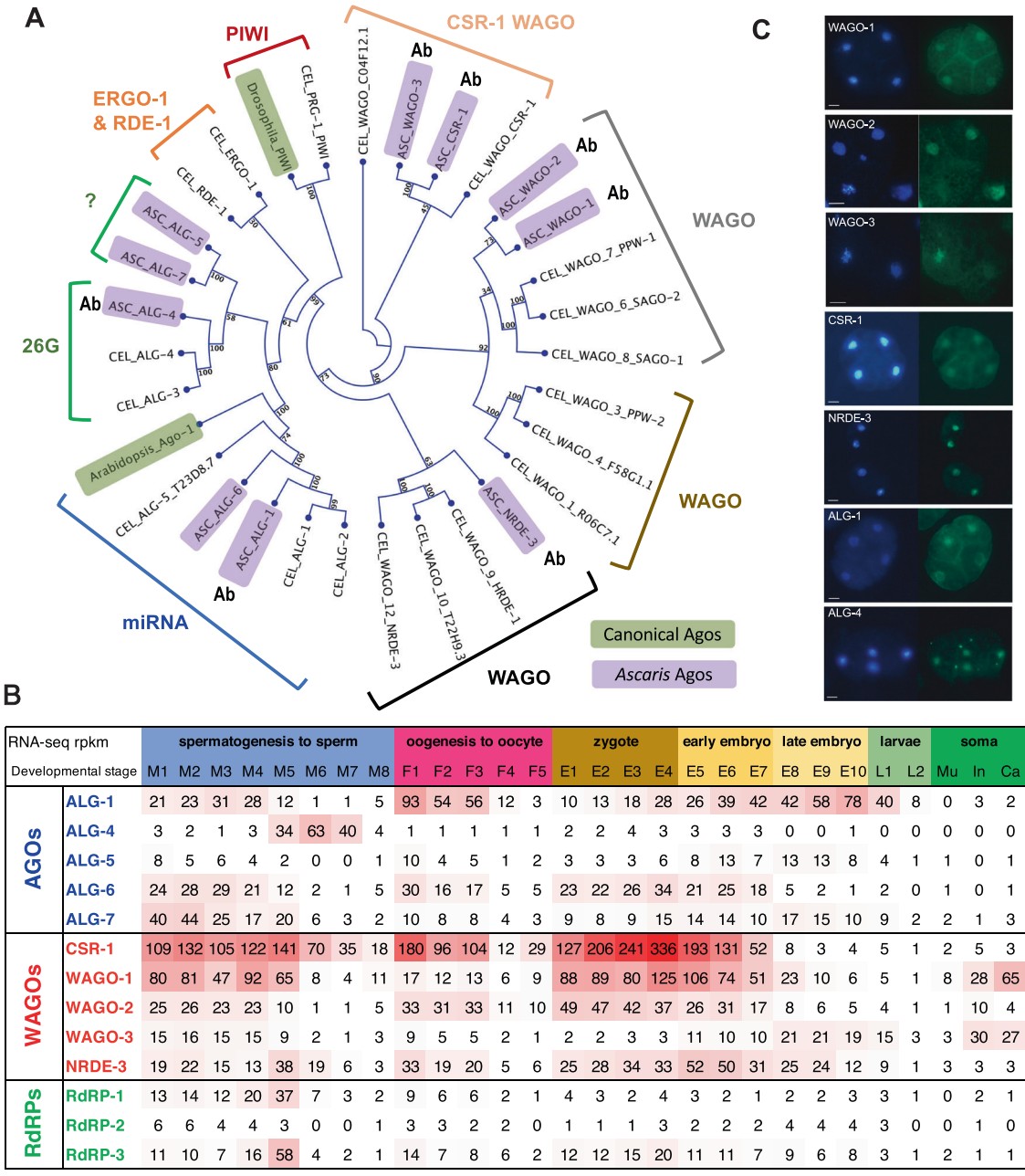

**Fig. 1 *Ascaris* has 10 Argonautes. A** Argonaute protein phylogenetic tree showing the relationship of *Ascaris* and *C. elegans* Argonautes. *Ascaris* Argonautes are in purple boxes. Labels indicate the type of Argonaute or its associated small RNA. GenBank accession numbers for these Argonaute protein sequences are available in Supplementary Table 1. Argonautes highlighted in green represent a canonical fly PIWI and *Arabidopsis* AGO Argonaute. The Argonautes were aligned using Muscle and the tree generated using Neighbor Joining and Maximum Likelihood (1,000 bootstraps) with the bootstrap values illustrated. *Ascaris* proteins for which antibodies were generated are marked with Ab. **B** RNA expression of *Ascaris* Argonautes and RNA-dependent RNA polymerases (RdRPs). The levels of expression (numbers are in rpkm) are shown in a heatmap with higher expression in red. Note that we view an rpkm value that is less than 5 as a background level of expression. The developmental stages are as follows: M1 to M8 are regions of the male germline, with M1 = early mitotic region, M2 = late mitotic region, M3 = transition zone, M4 = transition zone to early pachytene, M5 = pachytene, M6 = late pachytene, M7 = meiosis diplotene to diakinesis, and M8 = spermatids (see Fig. 3, text, and Materials and Methods for a detailed description of the male germline stages); F1–F5 are regions of the female germline, with F1 = mitotic region, F2 = early pachytene, F3 = late pachytene, F4 = diplotene, F5 = oocyte; zygote maturation stages prior to pronuclear fusion isolated from the uterus (E1–4; E4 is the stage passed from *Ascaris* and the host to the environment)[60]; embryo development (at 30 °C) stages, with E5 = 24 h (1-cell), E6 = 46 h (2-cell), E7 = 64 h (4-cell), E8 = 96 h (16-cell), E9 = 116 h (32–64-cell), E10 = 7day (256-cell), larvae L1 (10-day) and L2 (21-day)[60]; and adult somatic tissues, with Mu Muscle, In intestine, and Ca carcass, which includes the cuticle, hypodermis, muscle, nervous system, and pharynx. The RNA-seq expression data are provided as a Source Data file. **C** *Ascaris* Argonaute immunohistochemistry in early embryos. DAPI (blue) in left panel and immunohistochemistry (green) in right panel. Data are derived from three or more biological replicates. Scale Bars = 10 μm.

Two *C. elegans* WAGOs function in nuclear RNAi and are expressed in different stages, HRDE-1 (germline) and NRDE-3 (soma)[58,59]. Our phylogenetic analysis suggests one *Ascaris* WAGO, named AsNRDE-3, is related to these *C. elegans* nuclear WAGOs. It is expressed in zygotes and early embryos and presents both in the cytoplasm and nuclei of embryos. We note that with only one *Ascaris* nuclear WAGO, AsNRDE-3 may also serve functions similar to *C. elegans* HRDE-1. Only AsCSR-1 and AsWAGO-3 have the RNase H, catalytic tetrad residues that confer slicer activity.

*Ascaris* has three RdRPs (Figs. 1B and 4B). One is orthologous to *C. elegans* RRF-3 (RdRP-3) and is expressed in the germline, oocytes, zygotes, and during early development. In the male germline, RdRP-3 RNA expression is highest at M5 when the initiation of 26G-RNAs occurs and likely plays a major role in their biogenesis. The other two *Ascaris* RdRPs are likely orthologous to either *C. elegans* EGO-1 or RRF-1 and are expressed in the germlines (Supplementary Table 1). No ortholog of *C. elegans* RRF-2 appears present in *Ascaris*. Additional *Ascaris* orthologous proteins likely associated with small RNA pathways in *C. elegans* are provided in Supplementary Table 1.

*Ascaris* Argonautes are dynamically expressed throughout the male and female germlines, early development, larvae, and adult tissues (Fig. 1B). The most abundant Argonaute in all these stages is the *Ascaris* CSR-1 ortholog, AsCSR-1 (Fig. 1B), which is expressed in the male and female germlines, zygotes prior to pronuclear fusion (where the maternal to zygotic transition occurs)[60], and through the 4-cell stage of early development. This expression profile suggests a pivotal role of AsCSR-1 and its small RNAs in germline development, the maternal-zygotic transition and early embryogenesis. AsCSR-1 is not expressed in late embryos, larvae or somatic tissues and the gene becomes heavily marked with H3K9me3 in 32–64 cell embryos when its expression ceases (Supplementary Fig. 1). Two isoforms of CSR-1 protein are present in *C. elegans* that vary in the first exon and thus amino-terminus[61–64]. The *C. elegans* CSR-1a isoform with the extended amino-terminus is expressed during spermatogenesis[63]. RNA-seq, ISO-seq, or PRO-seq analysis in *Ascaris* did not identify RNAs that differ in exon 1 and the AsCSR-1 has the extended amino-terminal end similar to *C. elegans* CSR-1a. Other *Ascaris* WAGOs are also primarily expressed in germline and embryos, with AsWAGO-1 expression at higher levels than the others. Interestingly, AsWAGO-1 and AsWAGO-3 are also expressed in somatic tissues such as the intestine and carcass (cuticle, body wall muscle, hypodermis, and some nervous tissue). Their targets and functions in these tissues remain to be determined.

For AGO-clade *Ascaris* Argonautes, the miRNA Argonautes (AsALG-1 and AsALG-6) are expressed in both the male and female germline, early development, and larvae but are present at much lower levels in somatic tissues including the muscle, intestine, and carcass. AsALG-6 is eliminated during *Ascaris* programmed DNA elimination[65–67]. The 26G-RNA Argonaute, AsALG-4, is predominantly expressed in the male germline and highest during meiosis. AsALG-5 and AsALG-7 expression is in general low in the male and female germlines and early embryos through the comma stage, except for AsALG-7 in early spermatogenesis. The phylogenetic analysis in Fig. 1A suggests they may be potential 26G-like Argonautes (Fig. 1A). In some phylogenetic analyses, AsALG-5 and AsALG-7 cluster with *C. elegans* ERGO-1 or RDE-1. However, *C. elegans* ERGO-1 binds 26G-RNAs that are 2'-O-methylated by HENN-1 but *Ascaris* lacks HENN-1[51].

**Ascaris Argonaute antibodies**. To identify small RNAs associated with the *Ascaris* Argonautes, we generated polyclonal antibodies (see Material and Methods) to AsALG-1 (miRNA), AsALG-4 (26G-RNA), and all five WAGO Argonautes (AsCSR-1, AsNRDE-3, AsWAGO-1, AsWAGO-2, and AsWAGO-3). These Argonautes were chosen as they represent the diversity of small RNA pathways in *Ascaris*, are the most highly expressed, and appeared orthologous to major *C. elegans* Argonautes. Western blot analyses show these antibodies specifically recognize the corresponding Argonaute proteins (Supplementary Fig. 2A). In addition, RNA and Northern blot analyses of the AsALG-1 and AsWAGO-1 IP indicate significant enrichment for miRNAs and 22G-RNAs supporting the specificity of these antibodies (Supplementary Fig. 2B). Immunoprecipitation and proteomic analysis of several of the Argonautes further demonstrated the specificity of the antibodies for their respective Argonaute, including known associated proteins in *C. elegans*. Additional IP and small RNA sequencing data (see below) support the specificity of the antibodies.

**Localization of Ascaris Argonautes in early embryos and the germline**. Early embryo immunohistochemistry using the antibodies to *Ascaris* Argonautes indicates that AsCSR-1, AsWAGO-1, AsWAGO-2, AsWAGO-3, AsNRDE-3, AsALG-4, and AsALG-1 are present in both the cytoplasm and nucleus in early embryos (Fig. 1C). AsWAGO-2 and AsWAGO-3 localize to condensed mitotic chromosomes during anaphase of a programmed DNA elimination mitosis (Zagoskin, M.V., Wang, J., Veronezi, G.M.B., and Davis, R.E., manuscript in preparation). AsCSR-1 does not localize to condensed mitotic chromosomes. We further examined the nuclear and cytoplasmic localization of several Argonautes in early embryos using Western Blots (Supplementary Fig. 2A). AsCSR-1 and AsNRDE-3 are present in both the cytoplasm and nuclei, but significantly greater in the cytoplasm. Notably, AsALG-1 is also present in nuclei and is associated with miRNAs. However, AsWAGO-2 is predominantly in the nucleus whereas AsWAGO-1 is slightly higher in the cytoplasm compared to the nucleus.

*C. elegans* Argonautes localize to a variety of germ granules (P-granules, Mutator foci, Z granules, and SIMR foci) in the germline and early embryo[68]. We carried out *Ascaris* Argonaute antibody histochemistry on the male and female germlines and early embryo. Using a variety of conditions, we have not been able to observe germ granules in the germline or early embryo. Similarly, other antibodies to *Ascaris* proteins known to localize in germ granules in the early *C. elegans* embryo (Dcp1, Dcp2, DcpS, Cgh-1, eIF4E, and others) also do not identify discrete granules in *Ascaris* early embryos. While it is possible that optimal fixation and other conditions may not have been identified for *Ascaris* germ granules, we have carried out successful immunohistochemistry in *Ascaris* embryos[69]. It is possible that such granules and functional complexes are present, but do not form large enough complexes to be visualized. Interestingly, CSR-1 antibodies did not identify P-granules in early embryos of *C. briggsae*[64] and P-granules have not been examined in other nematodes.

**Ascaris Argonaute antibodies IP specific small RNAs**. *Ascaris* Argonaute IP and small RNA sequencing demonstrate that the Argonautes associate with distinct small RNA classes that are antisense to different elements in the genome (Fig. 2A). For all IP and small RNA sequencing experiments, we carried out two or more biological replicates. The Argonaute IPs and small RNAs obtained are highly reproducible and the small RNA sequencing per sample was over 30 million reads on average and often over 50 million (Supplementary Table 2). In previous work we used a variety of approaches including preparing small RNA libraries

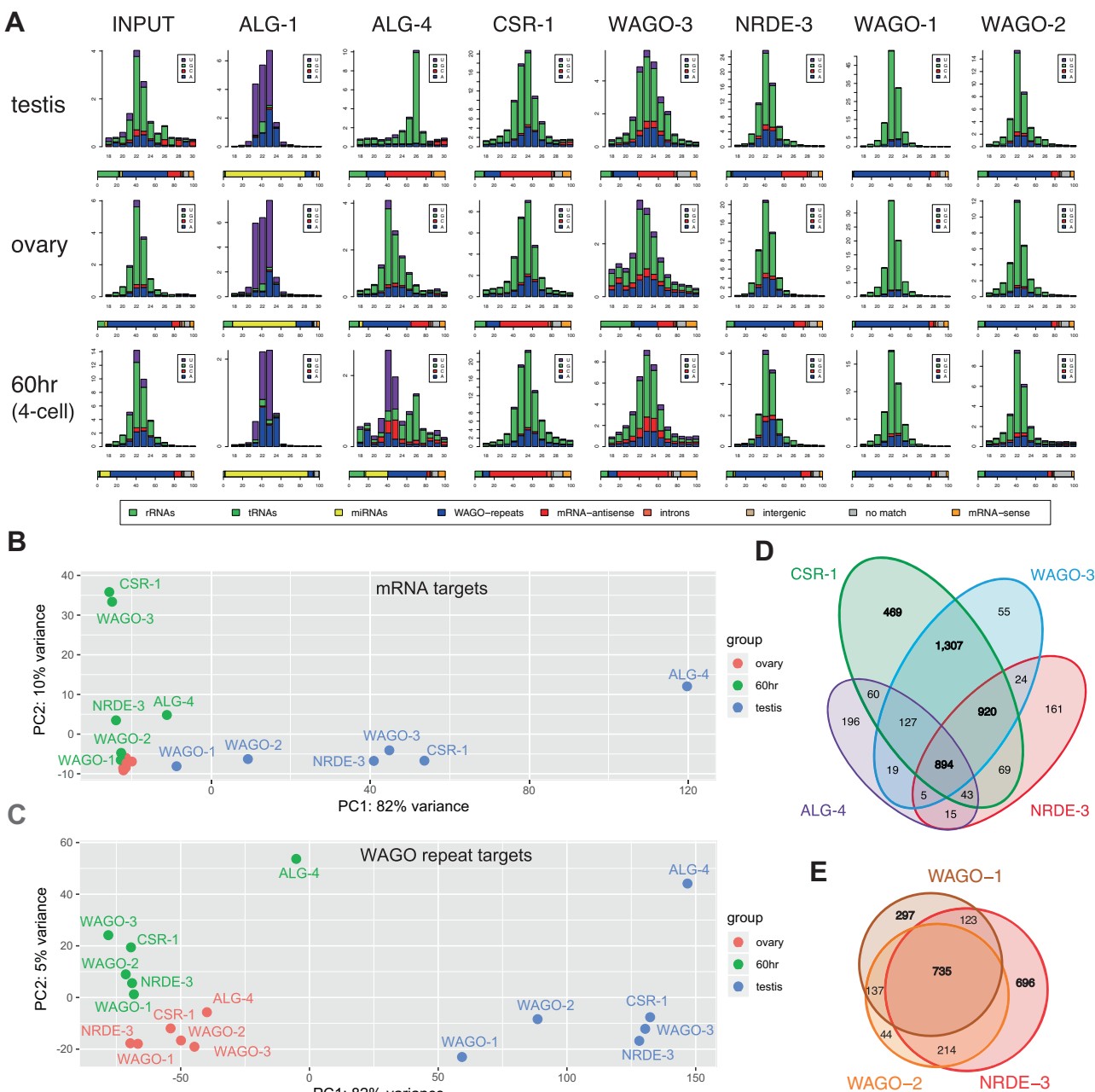

**Fig. 2 *Ascaris* Argonautes bind specific sets of small RNAs. A** Size distribution, frequency, and targets of small RNAs associated with specific Argonautes. Small RNAs (18–30 nt) from input and Argonaute immunoprecipitation (IP) were plotted for the whole testis (male germline), whole ovary (female germline), and 4-cell embryo (60 h). In these and all subsequent small RNA size distribution figures, small RNAs starting with A, C, G, and U of different sizes (*x*-axis) were plotted against their read frequencies (*y*-axis; raw reads in millions). Data are derived from two or more biological replicates combined. The small RNAs were categorized by their type (miRNA) or complementarity to different targets as shown in percentage bars below each size distribution plot. The target categories are sequences matching: (1) rRNAs or (2) tRNAs; (3) miRNAs; (4) repetitive sequences and mobile elements that are targets of AsWAGO-1, AsWAGO-2, and AsNRDE-3 (WAGO-repeats); (5) siRNAs antisense to mRNAs; siRNAs matching (6) introns or (7) intergenic regions; (8) small RNAs that have no full-length match to the genome (no match); (9) and sense small RNAs that correspond to mRNAs. **B**, **C** Principal component analysis (PCA) of small RNAs associated with Argonautes in different stages showing the overall relationship among these small RNA libraries. Panel **B** is PCA analysis of small RNAs that target mRNAs and **C** is small RNAs that target the WAGO-repeats. **D, E** Venn diagram showing the Argonaute small RNA overlapping targets in the testis for mRNAs (**D**) and WAGO-repeats (**E**). Source data are provided as a Source Data file.

that were both 5'-dependent and 5'-independent to determine if *Ascaris* small RNAs had 5'-monophosphate or 5' triphosphate termini[51]. In this study, most small RNA libraries were 5'-independent capturing all small RNAs regardless of their 5' phosphates (see Materials and Methods). However, AsALG-1 and AsALG-4 associated small RNA libraries were also constructed using 5'-dependent libraries to enrich for 5'-monophosphate

RNAs (see Supplementary Fig. 4). *Ascaris* AsALG-1 specifically binds 5'-monophosphate miRNAs (many previously identified in ref. [51]), AsALG-4 binds 5'-monophosphate 26G-RNAs, and the WAGOs typically bind 5'-triphosphate 22G-RNAs (22–24G-RNAs). Neither *Ascaris* miRNAs nor 26G-RNAs are represented at high abundance compared to the predominant 22G-RNAs (22–24G-RNAs) present in the total small RNA population

(see Fig. 2A, input). However, these small RNAs are highly enriched in AsALG-1 or AsALG-4 antibody IPs, respectively, demonstrating specificity of the Argonaute IPs (Fig. 2A). The majority of small RNAs in all *Ascaris* tissues and stages examined are the secondary, 5-triphosphate 22G-RNAs and they are primarily associated with AsWAGO-1 and AsWAGO-2. Much lower abundance 5-monophosphate 22G-RNAs are also present. Their relationship to the triphosphate 22G-RNAs could be as primary siRNAs or dephosphorylated triphosphate 22G-RNAs[70]. Argonaute IP from tissues where an Argonaute is expressed at low levels generates small RNA profiles reflecting the input or are random as seen in the ovary, and 4-cell IP of AsALG-4 (Fig. 2A) or AsWAGO-2 and AsWAGO-3 in the testis (Fig. 4, M6–M7), suggesting non-specific background for these IPs. Overall, the data indicate that the *Ascaris* antibodies are specific reagents for analysis of *Ascaris* Argonautes.

*Ascaris* chromosomes exhibit a relatively uniform distribution of genes[66] (Supplementary Fig. 5). However, repetitive sequences and mobile elements are biased toward DNA that is eliminated during programmed DNA elimination[65–67]. These eliminated sequences are at the ends of all germline chromosome (including 50–100 kb of subtelomeric repeat sequence that precedes the telomeres) and in the middle of a few chromosomes. The eliminated sequences include 30 Mb of a 120 bp tandem repeat that is present in the internal and terminal regions of chromosomes. Based on a liberal prediction, we estimate that repetitive and mobile sequences constitute ~41% or 126 Mb of the *Ascaris* germline genome (see Materials and Methods).

The majority of the *Ascaris* small RNAs (80–90%) are antisense to repetitive sequences and/or mobile elements in the genome (Fig. 2A, input and Supplementary Fig. 5). However, simple and satellite repeats such as the 120 bp tandem repeat (constituting 10% of the germline genome) are not targeted by small RNAs. Small RNAs to telomeric sequences have been observed in *C. elegans*[71], but no *Ascaris* small RNAs to telomeric sequences were observed in our in-depth sequencing (Supplementary Table 2). AsWAGO-1, AsWAGO-2, and AsNRDE-3 associated 22G-RNAs target the repetitive regions of the genome (5243 repeats high-confidence targets; see Methods and Source Data file). We hereafter refer to this repetitive sequence set as WAGO-repeats as they are targeted by *Ascaris* WAGO Argonautes. These WAGO-repeat regions are enriched with ≥10-fold small RNA reads (relative to mean coverage) and cover ~9.5 Mb or 3.4% of the genome (Source Data file and see also Fig. 5A, B). The majority (60–80%) of the AsWAGO-1, AsWAGO-2, and AsNRDE-3 associated small RNAs sequenced map to these WAGO-repeats (Fig. 2A, bottom horizontal percentage bar). AsWAGO-1 and AsWAGO-2 mRNAs are expressed at relatively similar levels and their small RNA targets largely overlap. AsNRDE-3 small RNAs and their targets show greater diversity compared to AsWAGO-1 and AsWAGO-2, particularly in the male germline during later meiotic stages where AsNRDE-3 associated small RNAs also target mRNAs (see below). The *Ascaris* genome has a larger content (40%) of defined repetitive sequences compared to *C. elegans*. However, the AsWAGO-1/2 targeting of repeats is not a consequence of the increased repeat content as less than 8% of these repeats (3% of the genome) are high-confidence AsWAGO-1/2 small RNAs targets.

To identify *Ascaris* genes that are potentially targeted by small RNAs, we normalized the small RNA reads to mRNAs based on reads-per-kb-per-million (rpkm). We considered mRNAs with antisense small RNAs over 10 rpkm in a stage as high-confidence targets (Source Data file). Small RNAs that target mRNAs are generally antisense and fully complementary to mature mRNAs and associated with AsCSR-1, AsWAGO-3, AsNRDE-3 (in the testis), and AsALG-4. The distribution of these small RNAs

across mRNAs is typically uniform across transcripts or with a 5' bias (Supplementary Fig. 6). These mRNA targeting Argonautes bind distinct sizes of small RNAs, with AsNRDE-3, AsWAGO-3, AsCSR-1 and AsALG-4 associated small RNAs 22G-, 23G-, 24G-, and 26G-RNAs, respectively (Supplementary Fig. 7).

AsCSR-1 small RNAs appear to target most expressed mRNAs and the levels are often proportional to mRNA target expression in most stages (see Supplementary Fig. 8 and Source Data file). These data are consistent with observations in *C. elegans* suggesting that AsCSR-1 could also function to "license" gene expression[24,26,27,62]. AsWAGO-3 is most likely orthologous to *C. elegans* C04F12.1. The targets of AsCSR-1 and AsWAGO-3 small RNAs are highly overlapping in many stages, particularly the 4-cell embryo (Fig. 2B and Source Data file). However, AsWAGO-3 is expressed at much lower levels compared to AsCSR-1 in germline and early embryos (Fig. 1B). Notably, *Ascaris* small RNAs corresponding to genes in general do not target pre-mRNA introns.

To further examine the diversity and complexity of small RNAs associated with different *Ascaris* Argonautes in the testis, ovary, and early embryo (4-cell), we compared the levels of small RNAs in these IPs matching mRNAs and repeats using principal component analysis (PCA) (Fig. 2B, C). Small RNAs to mRNAs are similar in the ovary, more divergent in the embryo, and highly divergent in the testes (Fig. 2B). Small RNAs to repetitive sequences associated with Argonautes show a similar relationship (Fig. 2C). These data indicate that small RNAs associated with *Ascaris* Argonautes are most diverse in the testes targeting both mRNAs and repeats. While AsCSR-1, AsWAGO-3, AsNRDE-3, and AsALG-4 small RNAs target different numbers of mRNAs in the testis (ALG-4 targets a much smaller number of mRNAs), many of these targeted mRNAs overlap (Fig. 2D and Source Data file). Repetitive sequences targeted by AsWAGO-1, AsWAGO-2, and AsNRDE-3 small RNAs also largely overlap, particularly in the testis (Fig. 2E and Source Data file).

**Dynamics of *Ascaris* Argonautes and their associated small RNAs during spermatogenesis.** Small RNAs and pathways have been extensively analyzed in the *C. elegans* female germline. In contrast, analysis of *C. elegans* small RNAs and pathways are more limited in the male germline[30–32,72–74]. 21U-RNAs, 26G-RNAs, and 22G-RNAs are all known to play an important role in the *C. elegans* male germline. Our PCA analysis of small RNAs suggested a greater diversity of small RNAs in the testis (Fig. 2B, C). *Ascaris* is sexually dimorphic and the extended male germline is ~1 m in length (Supplementary Fig. 3). This afforded us a unique opportunity to dissect and obtain large amounts of material from distinct regions of the testes, an attribute not available in *C. elegans*. We analyzed Argonautes, small RNAs, and target mRNA expression throughout *Ascaris* spermatogenesis to provide a comprehensive analysis of small RNA pathways during nematode spermatogenesis.

We defined five regions of the *Ascaris* male germline based on nuclear morphology and comparison with *C. elegans* (see Materials and Methods)[75,76]. These regions include the mitotic region, transition zone, early meiosis (pachytene), late meiosis I and II, and spermatids (Fig. 3). We dissected and collected seven samples of the male germline (M1–M7) that cover both mitotic and meiotic regions and carried out Argonaute IPs and small RNA sequencing as well as RNA-seq. Total small RNA and RNA-seq profiles were also generated for spermatids (M8) (Fig. 4C and Source Data file). The RNA-seq expression data indicates that most *Ascaris* Argonaute RNAs are significantly expressed during spermatogenesis (Fig. 4B) (except AsALG-5, see Fig. 1B). Overall, expression is highest during the early stages of spermatogenesis

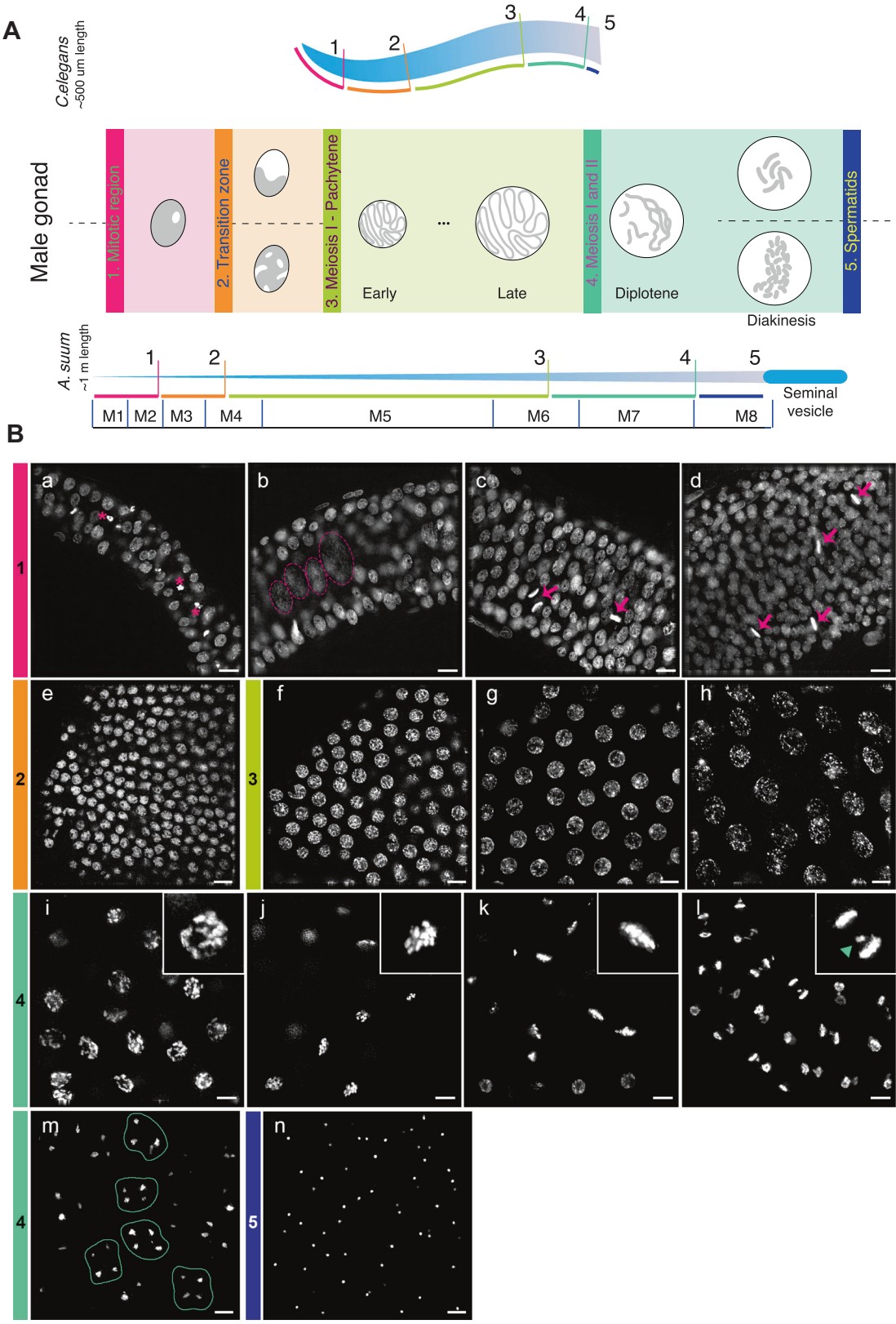

with expression decreasing during diplotene and diakinesis. The exception, however, is AsALG-4 with very low expression during early spermatogenesis and much higher expression during late meiosis (M6–M7).

The IP and small RNA sequencing data further illustrate the specificity of our Argonaute antibodies and reveal the types and targets of the small RNAs associated with each Argonaute during spermatogenesis (Fig. 4D). The size distribution of the small RNAs also reveals distinct features of many AsWAGO associated siRNAs in the testis (Supplementary Fig. 7). This suggests flexibility in the biogenesis and processing of these small RNAs or changes in loading or binding properties[77,78]. Notably, about

**Fig. 3 *Ascaris* male gonad regions and nuclear morphology. A** Schematic representation of regions of *Ascaris* male germline illustrated that correspond to *C. elegans* male germline and nuclear morphology. Corresponding regions between *C. elegans* and *Ascaris* are kept to scale; however, the length of the *Ascaris* male gonad is ~1 m (see Supplementary Fig. 3). Regions labeled are: (1) mitotic (pink), (2) transition zone (orange), (3) meiosis 1—pachytene, (4) meiosis 1 and 2 (green), and (5) spermatids (blue). Regions of the *Ascaris* male germline collected for Argonaute IP and RNA-seq are labeled M1–M8, separated by blue vertical lines. **B** Nuclear morphology of DAPI-stained regions of the *Ascaris* male gonad. Scale bars = 10 μm. **a–d** Overview of the mitotic region illustrating the progressive increase in gonad thickness. Mitotic stages are observed (metaphase, anaphase; arrows) and were confirmed by staining with CENP-A and the mitosis marker H3S10P, as well as apoptotic nuclei (*). Larger and less condensed nuclei are also detected, scattered through the mid-plane (dotted circles). **e** The transition zone exhibits more condensed and punctate nuclear morphology. **f–h** "Spaghetti bowl"-like nuclei characteristic of early (**f**), middle (**g**), and late pachytene (**h**). **i–m** Stages of meiotic progression: diplotene (**i**); diakinesis, with distinguishable individual bivalents (**j**); metaphase I (**k**); anaphase I (**l**), note lagging sex chromosomes (inset, arrowhead); and second meiotic division resulting in 4 haploid nuclei (**m**, contour). (**n**) spermatids. Data are derived from two or more biological replicates.

---

20–25% of all the AsCSR-1 small RNAs also begin with an adenine nucleotide. The low levels of AsWAGO-2 and AsWAGO-3 in M6–M7 result in poor IP and small RNA data. Similarly, the low levels of ALG-4 in early spermatogenesis result in small RNA IPs that look like input without distinct 26G-RNAs, whereas in M6–M7 where AsALG-4 is expressed, 26G-RNAs are clearly dominant in the IPs.

The targets of small RNAs associated with some Argonautes also change dramatically during spermatogenesis (Fig. 4). For example, AsNRDE-3 targets mainly WAGO-repeats in early stages (M1–M5) but then targets mRNAs in late meiosis (M6–M7). AsALG-4 associated 26G-RNAs only appear in late meiosis (M6–M7) targeting male, meiosis-specific mRNAs. AsWAGO-2 and AsWAGO-3 do not IP any specific small RNAs in the M6–M7. Interestingly, two RdRPs are also upregulated in the pachytene stage (M5, Fig. 4B). Overall, the data illustrate major Argonaute and small RNA changes during meiosis (later pachytene) and are consistent with a role for siRNAs in clearing most mRNAs prior to final sperm maturation (see below) as very few mRNAs are present in *Ascaris* spermatids[51,60].

**Repeat targets of small RNAs during spermatogenesis.** *Ascaris* WAGO-1, WAGO-2, and NRDE-3 associated small RNAs extensively target repetitive sequences including mobile elements (defined as WAGO-repeats) during spermatogenesis. Many of these targets overlap with enriched H3K9me2/3 histone marks (Fig. 5A, B). However, not all repetitive or mobile elements, H3K9me2/3 regions, and WAGO small RNA targets of the genome overlap. AsWAGO-2 and AsNRDE-3 associated repeat small RNAs decrease as spermatogenesis progresses, particularly in M6–M7 (Fig. 5). In the testes, WAGOs target a subset of repetitive sequences at high levels in the mitotic germline. Most of these repetitive sequences are not expressed and thus appear silenced (Fig. 5A and Source Data file). However, some WAGO-repeat regions are expressed, particular during pachytene (M4–M7, see Fig. 5C and Source Data file). Overall, small RNAs target many repetitive sequences or mobile elements during spermatogenesis.

Several previous studies defined *Ascaris* sequences with characteristics of mobile elements[79–82]. We observed significant RNA expression (based on RNA-seq and PRO-seq) in the male and female germline (but not the early embryo) for one of these, a non-LTR R4 retrotransposon that inserts into the ribosomal locus. R4 was previously described as mobile based on its presence in different genomic locations among individuals[80,82]. In contrast to the expression of R4, our RNA-seq analyses suggest that most other predicted mobile elements (previous work and current predictions) are not highly expressed. However, PRO-seq suggests many of these mobile element loci are bidirectionally transcribed at low levels. This bi-directional transcription could lead to dsRNA substrates for Dicer to generate primary siRNAs, leading to subsequent secondary siRNA generation to further

silence the loci. Consistent with this, high levels of AsWAGO-1, AsWAGO-2 and AsNRDE-3 associated 22G-RNAs are observed to target many of these elements.

**mRNA targets of small RNAs during spermatogenesis.** To define the timing and expression of Argonaute associated small RNAs and their mRNA targets during spermatogenesis, we first identified 5526 genes targeted by at least one Argonaute (AsCSR-1, AsALG-4, AsNRDE-3, and AsWAGO-3) with antisense small RNAs over 10 rpkm in one or more of the spermatogenesis stages (see Source Data file). These genes can be categorized into two broad groups based on their expression profiles. One group includes genes expressed throughout male germline development (M1–M7; 4363 genes) whereas the other group are genes specifically expressed during meiosis (M5–M7; 1163 genes, of which 232 are eliminated by programmed DNA elimination in early development[65–67]). Figure 6A shows an example of these two groups as neighboring genes on chromosome 1. The gene on the left (ag_00250, a chloride intracellular channel exc-4) is expressed throughout the male germline whereas the gene on the right (ag_00251, a gene encoding a hypothetical protein) is only expressed during meiosis. Both genes are targeted by AsCSR-1, and their mRNA levels are correlated with AsCSR-1 siRNA levels across spermatogenesis. In contrast, the meiosis-specific gene ag_00251 is expressed during pachytene (M5–M7) and targeted by AsNRDE-3 in M5–M7 and AsALG-4 associated 26G-RNAs in M6–M7. Notably, 26G-RNAs are not detected during most of the early pachytene (M5, see Fig. 4D).

Genome-wide analysis reveals AsCSR-1 small RNAs target a large set of genes, and the number of genes targeted varies across developmental stages. In contrast, AsALG-4 is expressed and targets a much smaller number of genes that are specifically expressed primarily during testis meiosis (Fig. 6B). While AsNRDE-3 does not target many mRNAs during early spermatogenesis (M1–M4), the number of mRNA targeted rises to a level similar to AsCSR-1 in late spermatogenesis (M5–M7) (Figs. 6B and 7B). In late meiosis (M6), the mRNA targets largely overlap between AsCSR-1 and AsNRDE-3 (Figs. 6C and S7). While most AsALG-4 mRNA targets are also targeted by AsCSR-1 and AsNRDE-3, AsCSR-1 and AsNRDE-3 target many mRNAs not targeted by AsALG-4 (Fig. 6C).

The expression changes of Argonaute siRNAs and their targeted mRNAs identified two large groups mRNAs that are differentially regulated by these siRNAs (Fig. 6D). Many of these AsCSR-1-only targets are differentially expressed during spermatogenesis including in the mitotic region and transition zone (Fig. 6D). In contrast, AsALG-4 and its small RNAs are specific to mRNAs expressed during meiosis (M5–M7) and notably levels of these mRNA targets decrease when the AsALG-4 associated 26G-RNAs are expressed (Fig. 6D). 26G-RNAs appear specific to mRNAs expressed during meiosis (~72%, 833 out of 1163 meiosis-specific genes) (Source Data file). In comparison, AsCSR-1 small

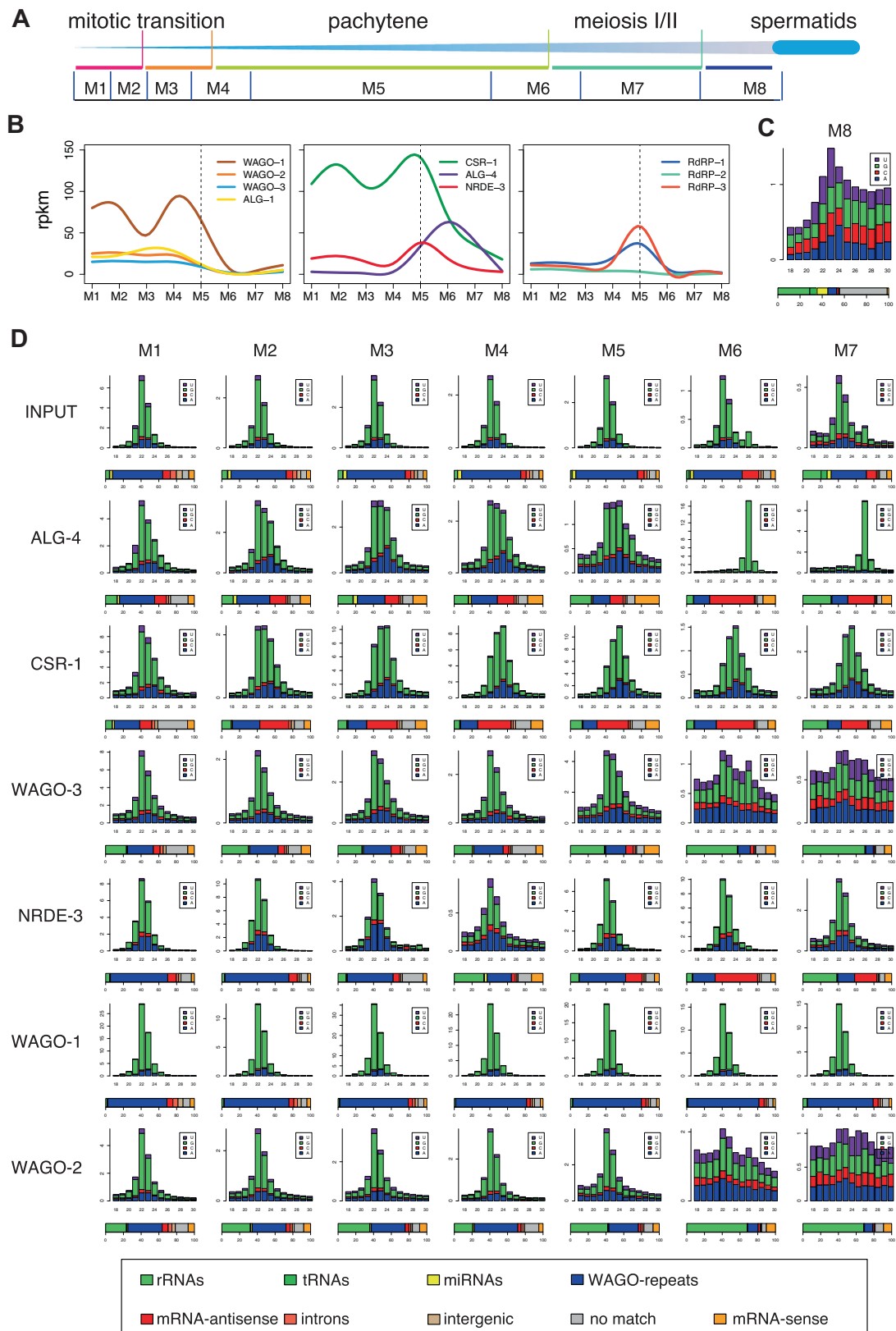

**Fig. 4 Small RNAs associated with *Ascaris* Argonautes during spermatogenesis. A** Illustration of *Ascaris* stages of spermatogenesis from Fig. 3. **B** RNA expression profiles of *Ascaris* Argonautes and RdRPs throughout spermatogenesis. The Argonautes are plotted in two groups based on their expression pattern or their targets. The RNA-seq expression data are provided as a Source Data file. **C** Small RNAs from mature spermatids (M8). The random size distribution pattern illustrates the low levels of bona fide small RNAs. **D** Small RNAs during spermatogenesis (M1–M7). In **C** and **D**, the size distribution, frequency, and classification of small RNAs are as described as in Fig. 2A.

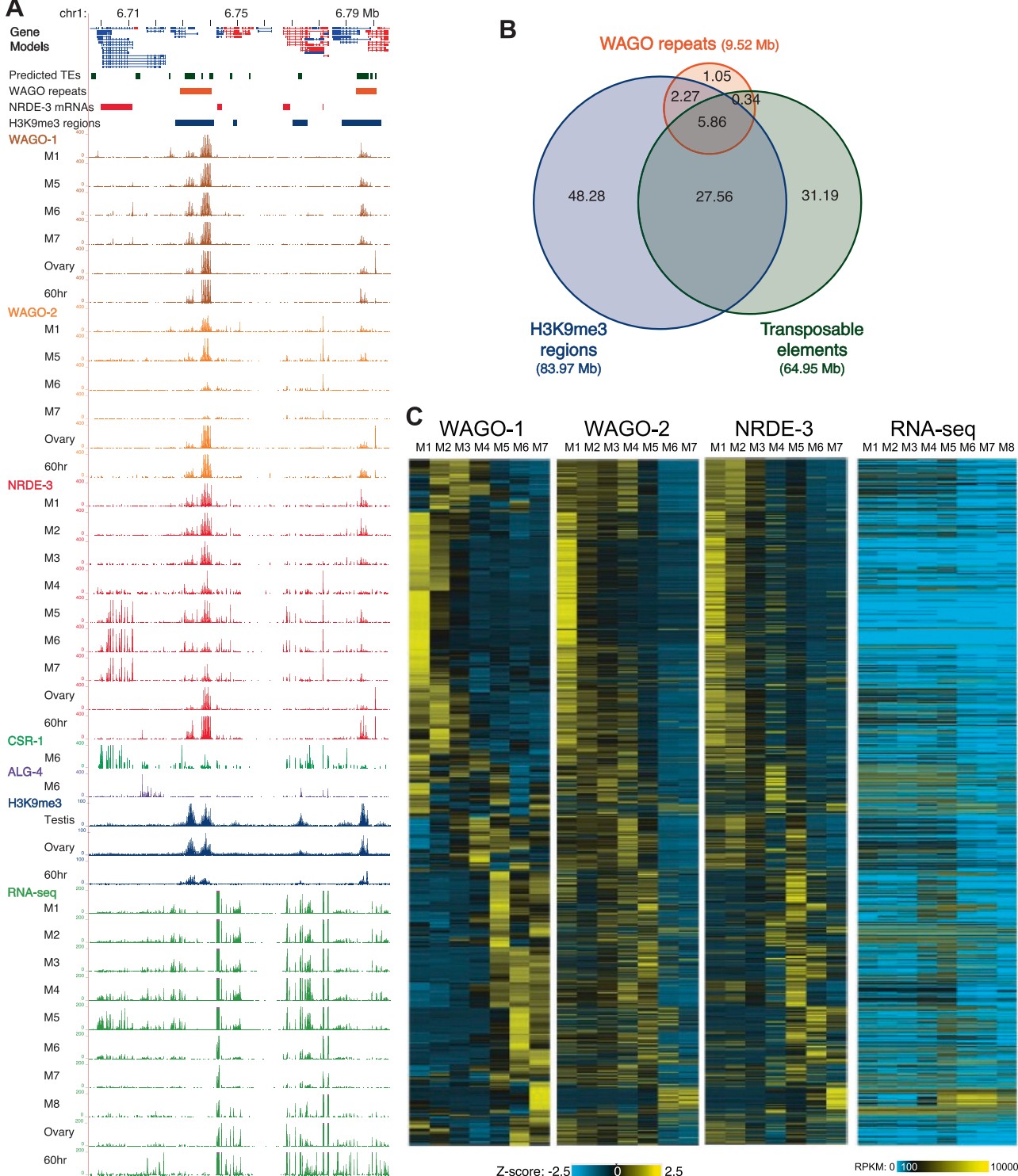

**Fig. 5 AsWAGO-1, AsWAGO-2, and AsNRDE-3 associated small RNAs target repeats during spermatogenesis. A** Genome browser view of a region of chromosome 1 illustrating Argonaute associated small RNAs, RNA expression (RNA-seq), and H3K9me3 levels. Only spermatogenesis stages that exhibit changes in expression or changes in Argonaute small RNA or mRNA expression are illustrated. Note NRDE-3 associated small RNAs change their targets from repeats to mRNAs during pachytene and late meiosis (M5–M7). **B** *Ascaris* WAGO-repeat targets largely overlap with genomic H3K9me3 levels and/ or the presence of transposable elements. **C** siRNA levels to the top targets (top 1776 WAGO-repeat loci; see Supplementary Table 3) for each WAGO during spermatogenesis. Heatmaps illustrate the standard Z-score (converted from rpkm) showing changes in expression of siRNAs associated with AsWAGO-1, AsWAGO-2, and AsNRDE-3. The RNA expression (rpkm) of the targets is also illustrated (right). Targeted loci were sorted based on the same order in the four heatmaps. Source data are provided as a Source Data file.

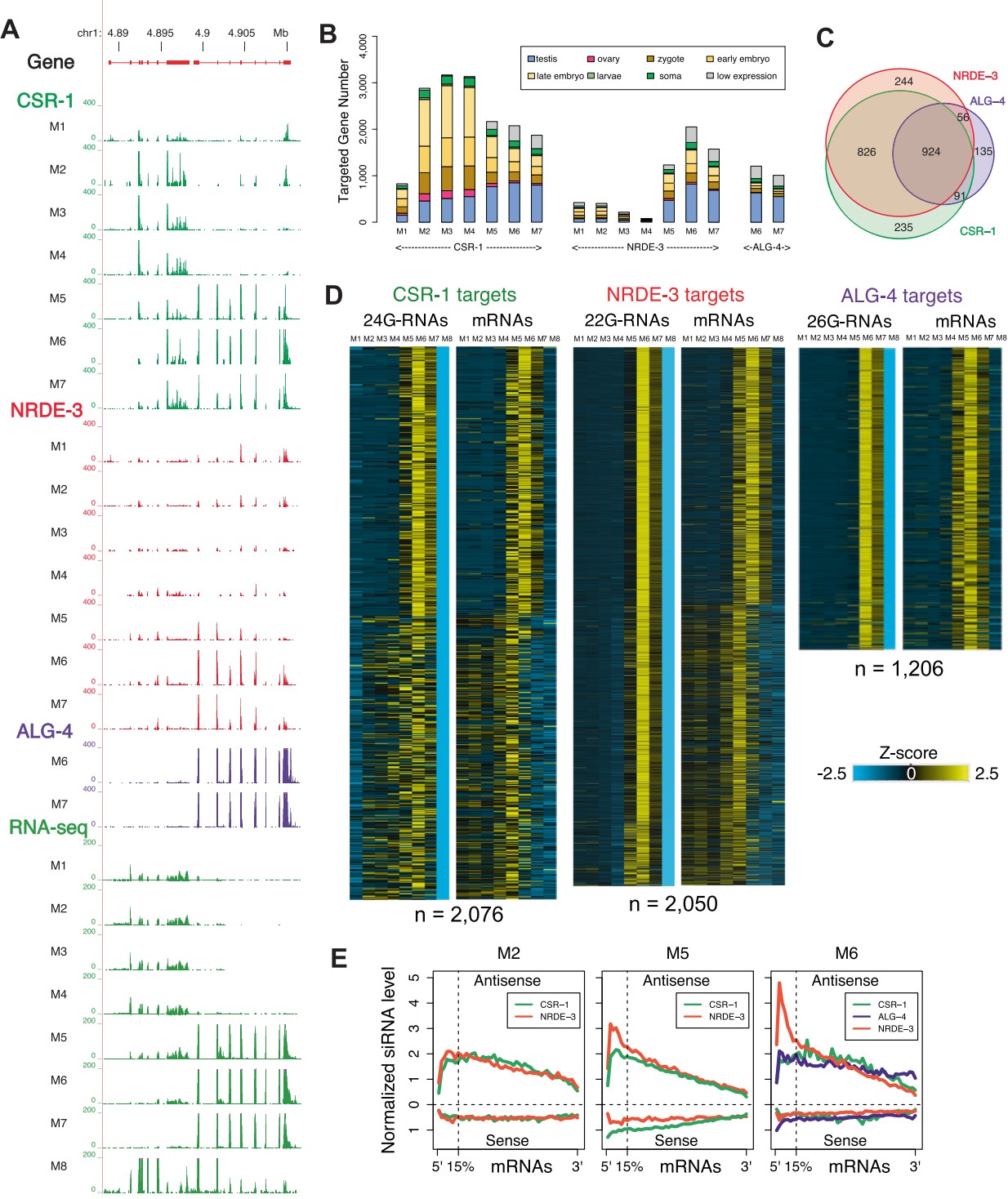

RNAs target most of the expressed mRNAs throughout spermatogenesis (~89%, 4910 out of 5526 expressed mRNAs) (Source Data file), but both the levels of AsCSR-1 and its small RNAs decrease significantly after M5. Notably, most AsALG-4 targeted mRNAs are expressed and also targeted by AsCSR-1 during pachytene (M5), but the corresponding 26G-RNAs are only expressed during later pachytene and meiosis (M6–7) (Fig. 6D). Overall, our data suggests a model where AsCSR-1 likely licenses, fine-tunes and represses the levels of all expressed mRNAs during early spermatogenesis and perhaps throughout spermatogenesis, while AsALG-4 targets and likely represses

meiosis-specific mRNAs. AsNRDE-3 small RNAs also target mRNAs later in meiosis, preferably at the 5' end of the mRNAs (Fig. 6E). Very low levels of mRNA are present in spermatids[51,60]. The decrease in AsCSR-1 levels during the later stages of spermatogenesis could result in a reduction in licensing or protection from repression facilitating a decrease in AsCSR-1 target mRNAs. Prior to the formation of spermatids, AsCSR-1 in concert with AsALG-4 and AsNRDE-3 small RNAs may also repress and facilitate the turnover of all mRNAs (Fig. 6D).

AsALG-4 targeted genes are predominantly male meiosis-specific genes. Many of these genes, including major sperm

**Fig. 6 AsCSR-1 and AsNRDE-3 license or fine-tune mRNA expression during early spermatogenesis whereas AsCSR-1, AsNRDE-3, and AsALG-4 facilitate the clearance of mRNAs during late spermatogenesis. A** Genome browser view of a region of chromosome 1 illustrating AsCSR-1, AsNRDE-3, and AsALG-4 associated small RNAs and their mRNA target expression during spermatogenesis. **B** AsCSR-1, AsNRDE-3, and AsALG-4 mRNA targets during spermatogenesis. AsCSR-1 small RNAs targets are pervasive in different stages and correlate with RNA expression, AsNRDE-3 small RNA targets switch from repeats to mRNAs in pachytene and late meiosis (M5–M7), whereas AsALG-4 is expressed and its associated 26G-RNAs target mRNAs that are primarily restricted to meiosis (M6–M7) in the male gonad. Shown are the number of mRNAs targeted by different Argonautes. The color indicates where these mRNA targets are most highly expressed in developmental stages using the same color scheme as Fig. 1B. Note AsCSR-1 and AsNRDE-3 target a broad group of mRNAs, while AsALG-4 targets mostly testis-specific genes. **C** AsCSR-1, AsNRDE-3, and AsALG-4 targeted mRNAs largely overlap during late meiosis. Venn diagrams showing the relationship between these targets in the M6 stage where all three Argonautes have a large number of targets. **D** siRNA and targeted mRNA levels in M6 (Fig. 6C) during spermatogenesis. Very low levels of small RNAs are in mature spermatids (Fig. 4C); thus, a 0 value is used in M8 for the AGO IPs. Heatmaps illustrate the standard Z-score (converted from rpkm) showing changes in siRNA and mRNA expression. Targeted genes were sorted based on the same order in the heatmap pairs. **E** AsNRDE-3 siRNAs are enriched for sequence at the 5′ ends of mRNAs. siRNA distribution on mRNAs changes in AsNRDE-3, AsCSR-1, and AsALG-4 during spermatogenesis (see also Supplementary Fig. 6). Source data are provided as a Source Data file.

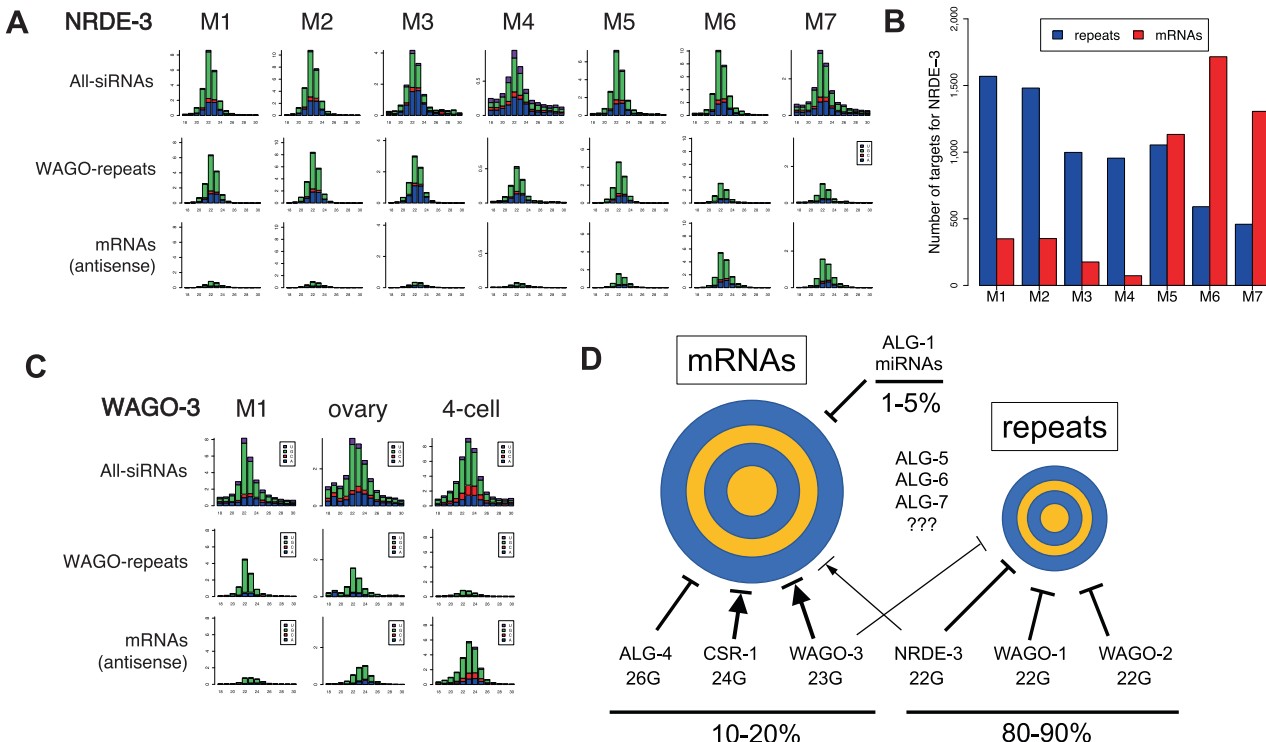

**Fig. 7 Plasticity of *Ascaris* small RNA pathways. A** AsNRDE-3 bound small RNAs change their targets from genomic repetitive sequence to mRNAs during spermatogenesis (M5–M7). **B** AsNRDE-3 mRNA and repeat target changes by during spermatogenesis. **C** WAGO-3 bound small RNAs change their targets from genomic repetitive sequence in male mitotic germline to mRNAs in 4-cell embryos. Source data are provided as a Source Data file. **D** Model for *Ascaris* small RNA pathways (see text). The two concentric circles represent *Ascaris* mRNAs (left) and repeats (right) targeted by small RNAs. The larger size of the mRNA circle indicates the larger number, complexity, and abundance of the mRNAs compared to the repeats targeted by small RNAs. Overall, however, in terms of abundance of small RNAs, 80–90% of all small RNAs target repeats. The 10 *Ascaris* Argonautes and their known associated major small RNAs (22 G, 23 G, 24 G, 26 G, and miRNAs) are shown. Lines with an arrow indicate licensing or fine-tuning (→) of expression, a blocking line indicates repression (—|), and an arrow with a blocking line indicates both licensing/tuning and repression (→|). Bolder lines indicate predicted stronger affects. Note several *Ascaris* Argonautes (CSR-1, WAGO-3, and NRDE-3) could be involved in both licensing and repression of their mRNA targets. In addition, NRDE-3 and WAGO-3 can target both mRNAs and repeats and their targets change through development, illustrating the plasticity of small RNA pathway in *Ascaris*.

protein genes, sperm motility genes, sperm-specific family genes, class P genes, and pp1 phosphatases are also targeted by *C. elegans* ALG-3/4[31,72] indicating conservation of 26G-RNA targets. However, many of the *Ascaris* 26G-RNA targets are unknown hypothetical genes and generally not conserved in other nematodes. New and novel genes have been shown to evolve at high levels in the testis[83–85]. The expression of these genes may require concerted control. AsALG-4 and its associated 26G-RNAs likely play a major role in clearing late meiosis mRNAs during

spermatid formation, as the timing of these 26G-RNAs expressions (M6–7) is in general later than their targeted mRNAs (M5–7), including AsALG-4 mRNA itself (Supplementary Fig. 9).

**Plasticity of small RNA pathways during spermatogenesis.** AsNRDE-3 largely targets repetitive sequences in the female germline, early embryo, and early stages of spermatogenesis. However, during the later stages of spermatogenesis, AsNRDE-3

small RNAs largely switch their targeting to mRNAs (Fig. 7A, B). The percentage of mRNAs targeted by AsNRDE-3 increases from ~10% in the mitotic regions of the male germline to 50% in late pachytene (M6, Figs. 4D and 6B). Over 2600 mRNAs are targeted by AsNRDE-3 during spermatogenesis, with 1716 mRNAs targeted alone in M6 (Fig. 7B). Most of these AsNRDE-3 targets overlap with AsCSR-1 targets (Fig. 6C, D, S8 and Source Data file). These data indicate plasticity in *Ascaris* NRDE-3 targets (repetitive sequences vs mRNAs) during spermatogenesis and suggest an important role for AsNRDE-3 during the later stages of spermatogenesis. This plasticity of small RNA targets is also observed in AsWAGO-3 (Fig. 7C). Although AsWAGO-3 expression is in general low, in germline tissues AsWAGO-3 targets WAGO-repeats while in early embryos it targets mostly mRNAs (Fig. 7C). Overall, these data indicate that plasticity in *Ascaris* Argonautes targets (Fig. 7D), with respect to the class of genomic elements, occurs in different developmental stages and tissues. These observations raise questions regarding the biogenesis of the small RNAs, their loading into the Argonautes, and how these processes change.

## Discussion

Our understanding of nematode small RNA pathways comes from detailed studies in *C. elegans*[8–12,68,86]. However, nematodes represent a very diverse phylum adapted to a wide variety of lifestyles with species present in almost all ecosystems including many parasitic species[37]. Little is known regarding the conservation and functional role of small RNA pathways (other than miRNAs) in these divergent nematodes[51,87,88]. Here, we characterize Argonautes and their small RNAs in the parasitic nematode *Ascaris*. *Ascaris* is a Clade III nematode that diverged from *C. elegans* (Clade V) ~365–400 million years ago[49,89]. *Ascaris* has a reduced set of expressed Argonautes compared to *C. elegans* (10 vs. 19) including 5 AGO-clade and 5 WAGO-clade Argonautes (Fig. 1A). *Ascaris* (as well as most non-Clade V nematodes) lacks piRNAs (21U-RNAs) and PIWI Argonautes[51,88]. We generated specific polyclonal antibodies to seven *Ascaris* Argonautes and analyzed small RNAs associated with AsALG-1, AsALG-4, AsWAGO-1, AsWAGO-2, AsWAGO-3, AsCSR-1, and AsNRDE-3 in early embryos, the female germline, and throughout the male germline. Overall, several small RNA pathways and their functions appear conserved between *Ascaris* and *C. elegans* while others have diverged in function and targets or been lost.

The most abundant small RNAs in *Ascaris* are 22G-RNA secondary siRNAs. The majority are antisense to repetitive sequences and are associated with *Ascaris* WAGOs, particularly AsWAGO-1 and AsWAGO-2. However, other *Ascaris* WAGOs, including CSR-1, WAGO-3, and NRDE-3, also associate with 22G-24G-RNAs that are antisense to mature mRNAs. Detailed analysis of AsCSR-1 and AsWAGO-3 small RNAs indicates they are primarily 24G-RNAs and 23G-RNAs, respectively (Supplementary Fig. 7), illustrating additional flexibility in average small RNAs sizes for their respective Argonautes. AsNRDE-3 and AsWAGO-3 small RNAs target mRNAs and/or repetitive sequences depending on the developmental stage. AsALG-1 binds miRNAs. AsALG-4 binds 26G-RNAs (like *C. elegans* testis ALG-3/4) that target mRNAs expressed primarily in the male germline during meiosis. AsALG-4 function may differ in *Ascaris* compared to *C. elegans*, whereas the AsCSR-1 pathways appear to function similarly as in *C. elegans*[24–27,62,90,91] even in the absence of an *Ascaris* piRNA pathway.

**A CSR-1 pathway in *Ascaris* in the absence of a piRNA pathway**. It has been suggested that a central role of piRNAs in

*C. elegans* is to define "self" vs "non-self" (*C. elegans* genes vs foreign elements)[56]. *C. elegans* piRNAs have limited, miRNA-like complementarity to their targets (seed sequences, nucleotides 2–8, and additional 3' sequences, nucleotides 14–19), thus providing the potential for targeting a vast array of sequences[53,92,93]. It has been proposed that a role of the *C. elegans* CSR-1 pathway is for "licensing" gene expression, thus enabling expression of genes, to counteract promiscuous piRNA silencing of germline transcripts[24,26,27]. Given the absence of piRNAs and PIWI in *Ascaris*, we sought to determine the targets of AsCSR-1 small RNAs and thus the potential function of the *Ascaris* CSR-1 pathway. AsCSR-1 represents the most abundant Argonaute mRNA in the *Ascaris* germline and early embryos. Its associated 24G-RNAs are antisense to almost all transcribed mature mRNAs. Why does AsCSR-1 target so many mRNAs? Does this indicate it still plays a role in "licensing" gene expression in *Ascaris*? In the absence of piRNAs, perhaps the *Ascaris* WAGOs or other genome regulatory processes involved in gene silencing (including programmed DNA elimination[94]) must be counteracted. Alternatively, the AsCSR-1 pathway may not be involved in "licensing gene expression". Instead, it functions in fine-tuning gene expression[90] or reducing noise to prevent epimutations[95].

*C. elegans* CSR-1 has been proposed to "fine tune" gene expression of mRNAs loaded into oocytes[90], clear maternal mRNAs[25], and degrade mRNAs with non-optimal codons[96]. AsCSR-1 is present in oocytes and at its highest level during the maternal to zygotic transition which initiates at the one-cell stage in *Ascaris* prior to pronuclear fusion[60]. AsCSR-1 disappears early in *Ascaris* development at the 32–64 cell stage. These data are consistent with AsCSR-1 also playing a similar role as observed in *C. elegans* in fine-tuning oocyte transcripts and clearing mRNAs during the maternal to zygotic transition. The balance among these functions of AsCSR-1 (licensing, fine-tuning, and repressive) may be developmental stage and/or mRNA dependent and remains to be determined even in *C. elegans*.

**Ascaris Argonautes and small RNAs in spermatogenesis**. Small RNA pathways and targets throughout the developmental stages of nematode spermatogenesis have not been analyzed. *Ascaris* is sexually dimorphic. We used the ~1 m length of the male *Ascaris* germline to comprehensively examine mRNA, Argonautes and their associated small RNAs, and the small RNA targets throughout nematode spermatogenesis. A striking feature of *Ascaris* small RNA pathways is the unique expression of AsALG-4 and associated 26G-RNAs during the later stages of spermatogenesis (M6–M7). This is followed by a significant decrease in the mRNAs targeted by the 26G-RNAs (Fig. 6). AsALG-4 and 26G-RNAs may be similar in function to pachytene piRNAs in mice that facilitate mRNA clearance in spermatids[97–100]. This function may be enhanced with AsCSR-1 and AsNRDE-3 that also appear to repress mRNAs during pachytene and later meiosis (see below). Thus, independently evolved small RNA pathways may have convergent functions and play key roles during pachytene of spermatogenesis.

In *C. elegans*, it has been proposed that 26G-RNAs likely identify spermatogenesis mRNA targets for generation of secondary 22G-RNAs[29–32,72]. This does not appear to be the case in *Ascaris* as our current data further refine our previous finding[51] and indicate that AsALG-4 and 26G-RNA expression does not precede the appearance of secondary 22G-RNAs to the same mRNA targets. In contrast, our data suggest that 26G-RNAs appear much later than most of the small RNAs targeting mRNAs during spermatogenesis and 26G-RNAs are associated with the loss of mRNAs in late spermatogenesis and the formation of spermatids (Fig. 6).

The current *Ascaris* data are consistent with a model where AsCSR-1 small RNAs license gene expression during early stages of spermatogenesis, the female germline, and early embryo similar to the licensing role of CSR-1 in *C. elegans*[24,26,27,62]. We show that AsALG-4 26G-RNAs appear during the late stages of spermatogenesis commensurate with when their target mRNAs disappear. AsALG-4 26G-RNAs are uniformly distributed across their target transcripts (current data and[51]). This contrasts with the 5' and 3' end targeting of transcripts of *C. elegans* ALG-3/4 26G-RNAs during spermatogenesis[72]. Thus, AsALG-4 transcript targeting differs from *C. elegans* ALG-3/4 indicating flexibility and differences in targeting between nematodes. The AsALG-4 transcript targeting appears more similar to *C. elegans* ERGO-1 26G-RNAs that uniformly target across transcripts[72].

Strikingly, AsNRDE-3 22G-RNA targets are largely repetitive sequences during early spermatogenesis but during the later stages (M6–M7) their targets are predominantly mRNAs. *C. elegans* NRDE-3 is involved in the nuclear RNAi pathway and contributes to co-transcriptional repression (inhibiting elongation)[59,101]. AsNRDE-3 small RNAs target the 5' ends of transcript in M6–M7. This may be associated with co-transcriptional repression of targets. The decrease in AsCSR-1 levels during the later stages of spermatogenesis may lead to loss of licensing or protection from repression also resulting in a decrease in AsCSR-1 target RNAs. AsCSR-1 could also play a repressive role as recently described[25,96]. Notably, targets of AsCSR-1, AsALG-4, and AsNRDE-3 in M6–M7 largely overlap. *Ascaris* spermatids exhibit very low levels of mRNAs and small RNAs. We suggest that during the later stages of *Ascaris* meiosis these Argonautes, their changes in expression, and small RNAs act in concert to clear mRNAs in the formation of spermatids. Overall, complex and diverse small RNA pathways contribute to key patterns of gene regulation likely involving licensing during early spermatogenesis and transcriptional and post-transcriptional repression during the later stages of *Ascaris* spermatogenesis.

**Plasticity of *Ascaris* WAGO-3 and NRDE-3 targets.** *Ascaris* WAGO-1 and WAGO-2 associated small RNAs only target repetitive sequences. However, AsWAGO-3 and AsNRDE-3 small RNAs, depending on the stage and tissue, can target either repetitive sequences or mature mRNAs. AsWAGO-3 small RNAs in early embryos predominantly target mature mRNAs that overlap with AsCSR-1 (AsWAGO-3 RNA is expressed at much lower levels than AsCSR-1) whereas in the male gonad they largely target repetitive sequences. Sequence alignment and phylogenetic analyses suggest AsWAGO-3 appears similar to *C. elegans* C04F12.1 and clusters with CSR-1. *C. elegans* NRDE-3 IP and 22G-RNA analysis indicate the small RNAs target intergenic repetitive loci[102]. In *Ascaris*, NRDE-3 also targets primarily repetitive sequences during early spermatogenesis, the female germline, and in 4-cell embryos. However, during the later stages of spermatogenesis, a large percentage of NRDE-3 small RNAs target mRNAs. Secondary 22G-siRNAs in *C. elegans* are generated by RNA-dependent RNA polymerase following identification of target RNAs by primary siRNAs. The change in small RNA targets in AsWAGO-3 and AsNRDE-3 raises questions regarding how distinct types of targets are identified as substrates for the biogenesis of small RNAs, the RdRPs used to generate the 22G-RNAs, and the loading of these RNAs into distinct Argonautes in different stages[77,78].

**_Ascaris_ primary siRNAs.** Several studies suggest that the introduction of dsRNA leads to RNA interference in *Ascaris* embryos (Davis, R.E., unpublished), larvae[103,104], the isolated and perfused adult intestine[105], and muscle[106] through injection of dsRNA into the pseudocoelom of adults. The key *C. elegans* RDE-1 Argonaute binds exogenous siRNAs derived from dsRNA. *Ascaris* appears to lack an RDE-1 Argonaute. What *Ascaris* Argonaute then acts on introduced dsRNA or endogenous dsRNA to generate primary siRNAs that induce the generation of secondary siRNAs? Most of our phylogenetic analyses suggest that AsALG-5 and AsALG-7 appear related *C. elegans* ALG-3/4, while in a few cases, they cluster with either RDE-1 or ERGO-1. AsALG-7 expression is highest in the early stages of spermatogenesis. However, we observe very low levels of 26G-RNAs in early spermatogenesis suggesting AsALG-7 may not bind 26G-RNAs. The lack of HENN-1 and 26G-RNAs in other stages suggests that AsALG-5 and AsALG-7 are not likely orthologs of ERGO-1. It is possible that the widely expressed AsALG-7 might serve functions similar to *C. elegans* RDE-1. Overall, with no *Ascaris* piRNAs and 26G-RNAs not likely serving as primary siRNAs to initiate secondary siRNA generation, the origin of endogenous primary siRNAs in *Ascaris* that lead to the abundant 22G-RNAs is unclear. Recent data suggest that in some *C. elegans* stages or compartments the CSR-1 slicer activity[22] may initially function to target and cleave mRNAs thereby serving to identify RNAs for RdRPs to initiate the generation of secondary siRNAs for CSR-1[96]. A similar mechanism may occur in *Ascaris*.

**Evolutionary comparison of Argonautes and small RNA pathways in nematodes.** A diverse array of repetitive sequences are targeted by 22G-RNAs in *Ascaris* and these are the most abundant small RNAs. Notably, *Ascaris* Argonautes that associate with small RNAs targeting repetitive sequences (AsWAGO-1, AsWAGO-2, and AsNRDE-3) do not have the catalytic tetrad for slicing activity.

Many mobile elements in *Ascaris* are transcribed bidirectionally at low levels and may generate dsRNAs for primary siRNA generation that may then lead to secondary, amplified 22G-RNAs and suppression of mobile element expression. High levels of these 22G-RNAs target mobile elements in most developmental stages. Many, but not all of these loci exhibit repressive histone marks including H3K9me2/3 (Fig. 5). The *Ascaris* genome lacks de novo DNA methyltransferases and cytosine DNA methylation appears absent in many nematodes[107,108]. Thus, a key role of *Ascaris* small RNAs and their pathways is likely to silence mobile elements and other repetitive sequences through repressive histone chromatin marks.

Other Clade III–IV nematodes also target mobile elements in the absence of piRNAs. Although 22G-RNAs appear absent in Clades I and II[88], other types of small RNAs are generated to mobile elements in these clades. These small RNAs are likely derived from RdRPs generating dsRNA that are cleaved by Dicer similar to plant and fungal systems that generate monophosphate siRNAs[2]. Cytosine DNA methylation is present in some Clade I nematodes[108–110] and siRNAs may lead to DNA methylation and transcriptional repression of mobile elements. Taken together, different nematodes appear to use diverse mechanisms to repress mobile elements[88].

Previous analysis of the evolutionary conservation of Argonautes and different types of small RNAs in nematodes suggest that the complexity of Argonautes and small RNA pathways in *Caenorhabditis* is not conserved throughout nematodes[8,40,41,51,87,88,107,111–113]. The piRNA pathway is present only in Clade V nematodes; *C. elegans* CSR-1 and/or C04F12.1 orthologs are present in some Clade III, V, and IV nematodes, but appear absent in Clade I and II nematodes; and nuclear Argonautes (HRDE-1 or NRDE-3) are present in many *Caenorhabditis* and some Clade III nematodes, but they appear

absent in other Clades[8,107,110,111,113]. Here, we have shown general conservation of *Caenorhabditid* CSR-1, ALG-3/4, and WAGO-associated small RNAs and their targeting in *Ascaris* (Clade III). Secondary siRNAs (22G-RNAs or longer) are present in Clades III–V, but they are absent in Clades I–II[8,87,88,112,113]. *Strongyloididae* (Clade IV) appear to have 27GA-RNAs instead of 22G-RNAs and the secondary siRNAs of *Globodera pallida* (Clade IV) are 22–26 nt in length[87,88]. Differences in the sizes of small RNAs is also observed in *Ascaris* with discrete small RNAs (22–24 G) associated with different WAGOs. Although orthologs of ALG-3/4 appear present in most nematodes, 26G-RNAs appear generally absent; *Ascaris* appears to be an exception with abundant 26G-RNAs during spermatogenesis meiosis. If 26G-RNAs in other nematodes are restricted to meiotic regions of the male germline as seen in *Ascaris*, they may not have been enriched in samples used in previous analyses of those nematodes.

**22G-RNAs and transgenerational inheritance**. Transgenerational inheritance in *C. elegans* is associated with piRNAs, 22G-RNAs, and histone H3K9me3 and H3K27me3 marks[11,35,86,114]. *C. elegans* HRDE-1 is important for the maintenance of inheritance over several generations whereas NRDE-3 appears limited to one generation. An interesting question is whether Argonautes and small RNA pathways are associated with transgenerational inheritance in *Ascaris* and other nematodes. Transgenerational inheritance in *C. elegans* has been associated with environmental information (odors, temperature, food, other stress, etc.), behavior, lifespan, and viral and bacterial pathogens[35,86,115–120]. In parasitic nematodes, transgenerational inheritance of information like these and particularly host immune responses would be clearly beneficial. For example, exosomes from the parasitic nematode, *Heligmosomoides polygyrus (bakeri)*, contain 22G-RNAs and a WAGO Argonaute, and can suppress Type 2 innate responses and eosinophilia[112,121]. Nematode Argonautes and small RNA pathways may play key roles in host-parasite interactions and communication among nematodes. As many of these nematode Argonautes and small RNA pathways are worm-specific and thus differ from their hosts, they could be potential targets for new therapies against parasitic nematodes.

In summary, nematodes are a diverse phylum including free-living and parasitic species. *Ascaris* lacks piRNAs but maintains a CSR-1 pathway with small RNAs that target most expressed mRNAs. Thus, AsCSR-1 may act in a "licensing" pathway as observed in *C. elegans* in the absence of a piRNA pathway as well as serving other tuning or repressive functions. *Ascaris* ALG-4 associated 26G-RNAs target male meiosis-specific genes commensurate with their mRNA degradation and do not appear to act as primary siRNAs for targeting and generating secondary 22G-RNAs during spermatogenesis. Several *Ascaris* Argonautes, including AsNRDE-3 and AsWAGO-3 targets, are stage dependent altering their targets between mRNAs and repeats during spermatogenesis and development. Our data significantly expand our understanding of the conservation, divergence, and flexibility of nematode Argonautes and small RNA pathways. Nematode Argonautes and small RNA pathways may have initially evolved for specific types of gene regulation leading to better fitness, but over time may have been co-opted for other functions in diverse nematodes.

## Methods

**Ascaris**. Collection of *Ascaris* tissues, sperm, and embryonation was as previously described[51,60]. Male and female germline tissues were washed in PBS, frozen whole or following dissection in liquid nitrogen, and stored at −80 °C (see Supplementary Fig. 3, male germline). Lysates from frozen total germline tissue or dissected male germline were prepared by first grinding the frozen samples in a mortar and pestle

in liquid nitrogen followed by homogenization in a metal dounce at 4 °C in 20 mM Tris–HCl pH 7.9, 75 mM NaCl, 0.5 mM EDTA, 0.85 mM DTT. Nuclei were lysed by addition of 20 volumes of 20 mM HEPES pH 7.6, 300 mM NaCl, 0.2 mM EDTA, 1 mM DTT, 7.5 mM $MgCl_2$, 1 M urea, 1% NP-40(v/v), 200 U of NxGen RNase Inhibitor (Lucigen), and protease inhibitor cocktail (Roche)[122]. Fresh *Ascaris* 4-cell embryos were decoated and directly homogenized in a metal dounce and nuclei lysed as described above. Lysates were then used for total RNA isolation or immunoprecipitation.

**Antibodies**. We generated polyclonal antibodies to *Ascaris* fusion proteins for AsALG-1, AsALG-4, AsCSR-1, AsWAGO-1, AsWAGO-2, AsWAGO-3, and AsNRDE-3 and also to peptides for AsWAGO-2 and AsWAGO-3. AsALG-1 amino acids 1-165 (HQ611964), AsALG-4 amino acids 1–75 (HQ611965), AsCSR-1 amino acids 1–162 (HQ611969), AsWAGO-1 amino acids 1–113 (HQ611970) AsWAGO-2 amino acids 1–141 (HQ611971), AsWAGO-3 amino acids 49–202 (HQ611972), and AsNRDE-3 amino acids 45–246 (HQ611973) were fused in frame with GST using the Bam H1 site of pGEX-6P-1 (GE Healthcare), the proteins were expressed in *E. coli*, purified using Glutathione Sepharose 4B columns, and used as the immunogens for the initial boost to generate polyclonal rabbit antibodies (Covance). For additional boosts, we used the *Ascaris* proteins cleaved from GST-fusions bound to Glutathione Sepharose 4B by treatment with Pre-Scission protease (GE Healthcare). AsWAGO-2 peptides (YASRRM-FELTKQSRDDYRA and RNGGQETSSSSSGEAGHLDI) and AsWAGO-3 peptides (VISYGRGKARRKEFPKQAGT and RENDLPRNGESVDWNRITD) were fused to KLH and used as immunogens to generate polyclonal rabbit antibodies (Pocono Rabbit Farm & Laboratory, Inc.). Antibodies were affinity-purified using either the peptides linked to SulfoLink beads (Pierce) or the *Ascaris* proteins (without GST) linked to a mix of Affigel-10/15 (BioRad).

**Embryo immunohistochemistry**. *Ascaris* embryo immunohistochemistry was carried out as described[60] using a modified freeze-crack method to permeabilize and fix embryos. Briefly, decoated embryos were suspended in 50% methanol and 2% formaldehyde solution and were frozen and thawed three times using a dry ice/ethanol bath. The embryos were re-hydrated with 25% methanol in PBS pH 7.4 for 1 min. After washing twice with PBS pH 7.4, the embryos were incubated in signal enhancer solution (Invitrogen I36933) for 30 min at RT. The embryos were then resuspended in blocking solution (0.5% BSA in PBS pH7.4) for 30 min at RT, followed by overnight incubation in primary antibodies at 4 °C, and then a 2 h incubation in secondary antibodies (Invitrogen) at room temperature. Nuclei were stained with DAPI.

**Cytological analysis of the *Ascaris* male germline**. Regions of 1–5 cm were collected from 1% formaldehyde-fixed dissected gonads at defined distances from the distal tip. Germline fragments were placed on a glass slide with PBS pH 7.4 and gently rolled over with a glass rod to release germline tissue from the outer somatic sheath. Samples were next stained with DAPI and the slides mounted in Prolong anti-fade medium (Invitrogen). Regions were characterized and defined based on nuclear morphology and presence or absence of mitotic/meiotic structures as follows: (1) Mitotic: round-shape interphase nuclei with evenly distributed chromatin and defined nucleolus and the presence of mitotic metaphases and anaphase. (2) Transition zone: more compact chromatin with punctate aspects and irregular shape, no mitoses observed. (3) Meiosis I—Pachytene: Chromatin aspects similar to a "bowl of spaghetti", increasing nuclear size from early (closer to TZ) to late (closer to meiotic progression). (4) Meiosis 1 and 2: increased chromatin compaction (diplotene), visualization of individual bivalents (diakinesis), further localization to the metaphase plate (metaphase I) and segregation to opposite poles with lagging material in the middle corresponding to sex chromosomes (anaphase I). While no clear observations of meiosis II prophase to anaphase were obtained, groups of 4 nuclei in very close proximity to each other indicated completion of the second meiotic division and formation of haploid nuclei. (5) Spermatids: dispersed, very small and compact nuclei.

Immunohistochemistry of the *Ascaris* germline was carried out on the same samples as described above. Tissue fragments were kept on the slide throughout the whole immunohistochemistry procedure, with careful pipetting of the solutions to avoid sample loss. Samples were first permeabilized in 0.2% Triton X-100/PBS for 15 min at room temperature, followed by two washes in PBS. Next, samples were suspended in blocking solution (1% BSA in PBS pH 7.4), covered with parafilm and incubated for 1 h at room temperature in a wet chamber. Primary and secondary antibody (Invitrogen) incubations were performed overnight at 4 C and for 2 h at RT, respectively, with slides covered with parafilm and placed in a wet chamber. After staining of nuclei with DAPI, slides were mounted in an anti-fade medium (Invitrogen) and kept in the dark at room temperature for 24 h before imaging.

**Image acquisition**. *Ascaris* germline immunohistochemistry and DAPI-stained preparations were imaged on an Applied Precision DeltaVision microscope, using a ×60 immersion objective and FITC/DAPI excitation filter set. Images were deconvolved with Applied Precision's Softworx software and analyzed using Fiji software. *Ascaris* embryos immunohistochemistry images were captured on a Leica

DM6B fluorescence microscope, equipped with a Leica DFC 7000 T camera, using a ×40 objective, FITC/DAPI excitation filter set and LASX software.

**Argonaute immunoprecipitation**. Argonaute IPs were carried out using affinity-purified rabbit antibodies pre-bound to Protein A Dynabeads (Fisher Scientific) (5–10 ug of antibody per 100 ul of beads)[123]. Germline or embryo whole-cell lysates were mixed with Protein A Dynabeads and rotated overnight at 4 °C. Protein A Dynabeads were recovered and washed with high-salt buffer (50 mM Tris–HCl, pH 7.4, 1 M NaCl, 1 mM EDTA, 1% Igepal CA-630, 0.1% SDS, 0.5% sodium deoxycholate) three times. The beads were resuspended in 250 ul of Proteinase K buffer containing 200 μg/ml Proteinase K and incubated for 1 h at 37 °C. RNA was extracted using Trizol LS (Invitrogen) adapted for small RNA recovery by precipitation using 2 volumes of isopropanol, 15 μg of GlycoBlue (Invitrogen), and 30 min incubation at −80 °C followed by centrifugation at $18,500 \times g$ for 35 min at 4 °C. The RNA was treated with RppH (25 units, New England Biolabs) in Thermopol buffer for 5' independent cloning[124]. Both untreated and RppH-treated samples were done for AsALG-1 and AsALG-4 samples. Following RppH treatment, samples were repurified with Trizol LS (Invitrogen) (adopted for small RNAs extraction) and stored at −80 °C.

**RNA, small RNA libraries, and sequencing**. RNA was isolated using Trizol or LS Trizol (Invitrogen). Total RNA samples were fractionated into small RNA (<200 nt) and long RNA (>200 nt) using the Monarch RNA cleaner (New England Biolabs). Small RNA libraries were prepared using the Small RNA-Seq Library Prep Kit (Lexogen). Both 5'-phosphate and 5'-phosphate independent libraries (RppH-treated)[124] were prepared. Libraries from 4-cell embryos were also prepared using 18–30 nt gel-purified RNA using the SMARTer smRNA-Seq Kit (Clontech) and NEXTflex-Small-RNA-Seq (New England Biolabs) with similar results. RNA > 200 nt was treated with TURBO DNase (Ambion) and rRNA depletion carried out using RiboCop rRNA Depletion Kit for Human/Mouse/Rat (Lexogen). Long RNA (>200 nt) libraries were made using CORALL Total RNA-Seq Library Prep Kit (Lexogen). Both small and long RNA libraries were sequenced 150 nt from both ends using the Illumina NovaSeq 6000 System.

**Small RNA data analysis**. Bioinformatic processing of small RNA sequencing data was carried out as previously described[51]. Briefly, adaptor sequences were trimmed and then the reads collapsed to non-redundant datasets. The reads were then mapped to the *Ascaris* genome and other various datasets using bowtie2[125] to determine targets and their expression levels. To define antisense regions in the genome corresponding to AsWAGO-1 and AsWAGO-2 small RNAs, we first mapped reads from each Argonaute IP-small RNA library to the genome using bowtie[126]. We normalized the coverage for each library to 30 million reads, a number close to the average number of raw input reads for our libraries. With an average length of ~22 nt for the small RNAs, this corresponds to ~2.5× coverage of the 280 Mb genome. As AsWAGO-1 and AsWAGO-2 small RNA targets are very similar, we merged genomic regions in close proximity (within 2,000 bp) if they have > = 20 fold average coverage (50×) from any of the 20 AsWAGO-1 and AsWAGO-2 libraries (using bedtools -mergeBed -d 2000). This merge is necessary as a locus is often not fully covered with high-levels of siRNAs. This approach defined an initial set of enriched loci (9164 with length ≥ 50 bp). We then used the normalized reads (rpkm) mapped to these loci and a 10-fold enrichment in at least one library to define these genome regions as enriched for AsWAGO-1 and AsWAGO-2 small RNAs. These regions were further filtered to remove loci related to rRNA, tRNAs, miRNAs or mitochondrial DNA. Overall, we defined 3948 AsWAGO-1 and AsWAGO-2 targeted (10× enriched) genomic regions that constitute 8.1 Mb of sequence and named them WAGO-repeats. Using the same approach, we defined 2912 (6.2 Mb) AsNRDE-3 targets. The majority (~77%) of the sequences defined as AsNRDE-3 targets overlap with AsWAGO-1/2 targets. See Source Data file for the defined WAGO-repeats and mRNAs and their WAGO-associated small RNAs. PCA analysis on the small RNA libraries was done using DESeq2 plotPCA[127]. Heatmaps were generated using Treeview 3[128].

**Repetitive sequence identification**. Repetitive sequences were identified using a combination of homology-based and de novo approaches, including RepeatMasker[129], LTRharvest[130], RepeatScout[131], RepeatExplorer[132], dnaPipeTE[133], MGEScan-non-LTR[134], Helsearch[135], MITE-Hunter[136], SINEfinder[137], TEdenovo[138], and RepARK[139]. The final collection of repetitive sequences was filtered for redundancy with CD-hit and are available within the repeat track at http://genome.ucsc.edu/s/jianbinwang/Ascaris_small_RNAs.

**Reporting summary**. Further information on research design is available in the Nature Research Reporting Summary linked to this article.

## Data availability
The small RNA and RNA sequencing data are deposited to NCBI GEO database (GSE189061). The data are also available in UCSC Genome Browser track data hubs[140] that can be accessed with this link: http://genome.ucsc.edu/s/jianbinwang/Ascaris_small_RNAs. Source data are provided with this paper, including small RNA

expression from INPUT and Argonautes IPs, as well as RNA-seq expression data for *Ascaris* genes and repeats that are targeted by these small RNAs. The data supporting the findings of this study are available from the corresponding authors upon reasonable request. Source data are provided with this paper.

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

## Acknowledgements

We thank Richard Komuniecki, Bruce Bamber, Jeff Myers, and Routh Packing Co. for their support and hospitality in collecting *Ascaris* material. We thank Adam Wallace and Stella Kratzer for initial work on the purification, evaluation, and western blot analysis of the *Ascaris* Argonaute antibodies. We thank Diane Shakes and Diana Chu for advice and suggestions for the analysis of *Ascaris* spermatogenesis and members of the *C. elegans* small RNA research community for comments and feedback on the manuscript. This work was supported by NIH grants to R.E.D. (AI049558 and AI114054) and J.W. (AI155588) and startup funds from the University of Tennessee at Knoxville to J.W.

## Author contributions

M.V.Z., J.W., and R.E.D. designed the project; R.E.D. performed Argonaute phylogenetic analysis; R.E.D. designed and prepared fusion proteins and peptides for *Ascaris* Argonaute antibodies; J.W. and A.T.N. characterized *Ascaris* Argonaute antibodies; A.T.N. carried out initial Argonaute immunoprecipitation and small RNA sequencing on embryos and M.V.Z. carried out comprehensive Argonaute immunoprecipitation and small RNA sequencing on embryos and the germlines; G.M.B.V. carried out cytological analysis of the *Ascaris* male germline with the help of J.W. and M.V.Z.; J.W. analyzed small RNA and RNA-seq data; M.V.Z. identified repetitive sequences; and J.W. and R.E.D. wrote the manuscript.

## Competing interests

The authors declare no competing interests.
