## [Peer Review File · Nature Communications]

Title: Small RNA Pathways in the Nematode *Ascaris* in the Absence of piRNAsREVIEWER COMMENTS

Reviewer #1 (Remarks to the Author):

This study is a tour de force characterization of the small RNA-Argonaute interactome of *Ascaris*. The results demonstrate the tremendous diversification of small RNAs and their Argonaute partners and reveal surprising fluidity in the targets of WAGO Argonautes during development. The results will provide a valuable resource to the small RNA field and will be instrumental in further explorations of *Ascaris* biology.

I'm a little confused by the AGO clade naming system used - I see the reason for trying to keep the names consistent with *C. elegans*, which is presumably why there is not an AsALG-2 or AsALG-3, but why then is there an AsALG-5 since it does not appear to be closely related to CeALG-5 (an Argonaute dear to my heart)?

Tai Montgomery

Reviewer #2 (Remarks to the Author):

Small RNAs are a crucial arm in genome defence across eukaryotes. Although deeply conserved, the specific nature of small RNA pathways diverges rapidly. Nematodes represent an excellent example of this because the piRNA pathway, thought to be essential for fertility and piRNA control in flies, mice and the nematode *C. elegans*, has been lost a number of times independently in different lineages across the phylum. Characterising other nematode small RNA pathways to better understand how piRNAs are replaced is really important and will give great insight into the essential core functions of piRNAs. In this paper the authors characterise small RNAs in the nematode *Ascaris* in great detail, using sophisticated molecular biology approaches to take this far beyond simple small RNA sequencing. It is an excellent resource for the community and makes some interesting points which could spark further research to test the hypotheses raised. I'm very enthusiastic about this study but I do have a few important comments about the interpretation and presentation of data in the manuscript, which are detailed below.

1. Title: this is far too general. Small RNAs evolve fast so the title "Nematode small RNA pathways in the absence of piRNAs" is misleading as different solutions will no doubt be found in other nematodes from different lineages.

I suggest that *Ascaris* is incorporated into the title, for example:

Small RNAs in *Ascaris*: genome regulation in the absence of piRNAs

Or something along these lines.

2. In the discussion, the observation that CSR-1 binds small RNAs antisense to a large number of protein-

coding genes is interpreted in support of a licensing function for CSR1. The authors need to add a discussion here of what they suppose CSR1 might be licensing against (since there are no piRNAs). They could suggest that the WAGO pathway is able to target genes in the absence of piRNAs and silence them, and that CSR1 protects this from happening. In addition I think that the authors should add a sentence saying that an alternative possibility is that CSR1 does not function in licensing at all and has an alternative function, such as fine-tuning gene expression or reducing noise in specific small RNA loci to prevent epimutations, as has been proposed in other studies.

3. The fact that the *Ascaris* Wagos target repeats is really interesting but needs a little more exploration in the text of the manuscript. In particular, can the authors quote in the manuscript the number of repeat targets compared to the number of protein coding gene targets, for all the IPs (including CSR1) and then carry out a statistical test of the enrichment for either repeats or protein coding genes in each case depending on the number of base pairs of protein coding genes and repeats respectively. I.e. if the genome is made up of 40% repeats, 10% protein-coding genes, and the wago targets are 99% repeats then this would likely be a significant enrichment using chi squared tests or Fisher's exact test (80%, as total bp that are available is repeats+protein coding genes, vs 99%). Additionally, it would be helpful if the authors could list some examples of the few genes that are targeted by wago and make some comment about what these are (they may be misannotated repeats). Finally it would be very useful for the authors to test whether there is still enrichment for repeats when considering only genes and repeats that are within H3K9me2/3 chromatin regions- is the apparent enrichment for repeats just because the elements within these regions are vastly more likely to be repeats?

These analyses are really important. If they hold and the enrichment for repeats is still very significant, then the authors need to point out in the discussion that this is different from *C. elegans*. My suspicion is that the authors will find that the enrichment, particularly when controlling for chromatin regions, is not significant and so the difference to *C. elegans* may be simply due to the fact that the *C. elegans* genome is a lot less repetitive; if this is the case then the authors would have to make that point instead.

Minor points

1. "Genome-wide analysis reveals AsCSR-1 small RNAs target a large, comprehensive set of genes"

The word comprehensive doesn't make sense here- please remove.

2. Can the authors replace "complementary" to antisense when they mean small RNAs that are antisense to particular targets? Complementary is ambiguous whereas antisense I feel is more direct.

3. In the discussion, the WAGO targets are described as "exclusively" repeats. This needs to be reevaluated in the context of the findings from the reanalyses described in point 3 above, in case they show something slightly different here.

Reviewer #3 (Remarks to the Author):

The manuscript by Zagoskin et al., describes small RNA pathways in the parasitic nematode *Ascaris* and

represents the most in-depth analysis of Argonautes and their small RNAs in a nematode (outside of the free-living nematode *C.elegans*). The data reported are substantial and incredibly high quality. The *Ascaris* model allows for analysis of Argonautes over many developmental stages and provides the first high resolution analysis during spermatogenesis, where the authors show regulation of AsALG-4 and its sRNAs (unique expression in M6-M7) and that AsALG-4 appears to repress specific mRNAs during meiosis in the testes. Another conclusion is that siRNAs aid in clearing mRNAs prior to final sperm maturation. The work is of broad interest to understanding how nematodes use small RNAs in development (and potentially pathogenesis) but also provides one of the most comprehensive set of data in any animal system. The authors show for example that *Ascaris* WAGOs can change their targets between repetitive sequences and mRNAs in different developmental stages – providing a really interesting and unique (in the literature) example of plasticity.

The challenge in reading this manuscript is just how much data there are – I believe it will continue to provide insights to others in the field (and be cited) for many years. The broad conclusions are that small RNAs are dynamic and flexible across development and also that there is divergence of nematode small RNA pathways between *C.elegans* and *Ascaris*. One of the questions posed by the authors is how *Ascaris* copes with a lack of piRNA pathway and whether small RNAs are still involved in licensing gene expression in this case. The authors use their AsCSR-1 IP data across different developmental stages, and paired with RNAseq data, to suggest that CSR-1 is involved in licensing gene expression in *Ascaris* and draw other parallels with *C.elegans*.

Comments/suggestions

Citation needed: “... nuclear Argonautes (HRDE-1 or NRDE-3) are present in many Caenorhabditids and some Clade III nematodes, but they appear absent in other Clades.

Citation needed: “...Secondary siRNAs (22G-RNAs or longer) are present in Clades III-V, but they are absent in Clades I-II.”

It would be worth a table/mention of the % identity of each *Ascaris* Argonaute to the *C.elegans* orthologs they are named after.

Figures

Figure 1 - From the phylogeny, why is CO4F12.1 called the CSR-1 clade? And how do the authors claim “AsWAGO-3 appears orthologous to *C. elegans* CO4F12.1 based on the phylogenetic analysis”

Figure 7 – A: the claim from this figure is that “AsNRDE-3 bound small RNAs change their targets from genomic repetitive sequences to mRNAs during spermatogenesis. However showing the first nt plots this way does not really show quantitative changes. I think an additional plot, like one shown in figure 6 with the proportion of each class of RNA/genome region, would help.

Figure 7 C- The schematic is a hard to follow (also not clear what the arrows mean in relation to the inhibition signal?).

Supplementary figures

In general a bit more information in the figure legends or annotation/labelling in the axes of graphs is merited.

Sfig 1: it might be worth better defining/annotating CSR-1 on this figure – what are all of the gene models?

Sfig 2: From these data - I do not draw the same conclusions on nuclear/cytoplasmic relations as the authors. From these blots, WAGO-3 is not shown and WAGO-2 seems only to be found in nucleus. But the authors state “AsWAGO-3, AsCSR-1, and AsNRDE-3 are present in both the cytoplasm and nuclei, but significantly greater in the cytoplasm” “WAGO-2 is equally present in the cytoplasm and nucleus whereas AsWAGO-1 is around two-fold higher in the cytoplasm compared to the nucleus.”?

Sfig 4: can the authors comment on why more ribosomal in testis region 4?

Sfig 8: More legend would help - for example what “RNA” in (A) on Y axis is referring to and how the data are normalized.

Sfig 9: It may be worth clarifying if the Y axis (after the top, mRNA) that simply say the AGO protein name are in fact small RNA reads from IP datasets – in general it should not be assumed that readers can jump in and follow at the level of the authors.

Methods

“We normalized the coverage for each library to 30 million reads, a number close to the average number of raw input reads for the libraries”. Slightly unclear: does this mean they subsampled the data or how exactly is this justified?

Additional

Some of the conclusions about WAGO expression and small RNA expression would be supported by a bit more detail: in the context of potentially different starting material amounts or affinity of antibodies, is there a quantitative way to claim that AsWAGO-1 and its small RNAs are expressed at much higher levels than AsWAGO-2?

Point-by-point Response to Reviewers

We thank the reviewers for their overall positive comments. We have addressed each of their questions and concerns below.

Reviewer comments are italicized in light grey with our responses non-italicized in black.

Reviewer #1

Why then is there an AsALG-5 since it does not appear to be closely related to CeALG-5?

The reviewer is correct that the AsALG-5 does not appear closely related to CeALG-5. However, we simply used a numerical naming, 1-7, after our 2011 small RNA work was published (<https://pubmed.ncbi.nlm.nih.gov/21685128/>). We skipped AsALG-2 and avoided AsALG-3 since *Ascaris* has only one Argonaute (AsALG-4) that matches CeALG-3 and CeALG-4 which both associated with testis 26G-RNAs. At the time, CeALG-5 has not been named. To clarify this, we have added a note in the text that AsALG-5 is not related to CeALG-5 in the first paragraph of the results “.....(Note: AsALG-5 is not orthologous to *C. elegans* ALG-5).”

Reviewer #2

1. Title: this is far too general. Small RNAs evolve fast so the title “Nematode small RNA pathways in the absence of piRNAs” is misleading as different solutions will no doubt be found in other nematodes from different lineages.

I suggest that Ascaris is incorporated into the title, for example:

Small RNAs in Ascaris: genome regulation in the absence of piRNAs

Or something along these lines.

The title has been changed to

“Small RNA Pathways in the Nematode *Ascaris* in the Absence of piRNAs”

2. In the discussion, the observation that CSR-1 binds small RNAs antisense to a large number of protein-coding genes is interpreted in support of a licensing function for CSR1. The authors need to add a discussion here of what they suppose CSR1 might be licensing against (since there are no piRNAs). They could suggest that the WAGO pathway is able to target genes in the absence of piRNAs and silence them, and that CSR1 protects this from happening. In addition I think that the authors should add a sentence saying that an alternative possibility is that CSR1 does not function in licencing at all and has an alternative function, such as fine-tuning gene expression or reducing noise in specific small RNA loci to prevent epimutations, as has been proposed in other studies.

We agree with the reviewer. We appreciate the comments including the idea of reducing noise in specific RNA loci to prevent epimutation. We have altered the text in the discussion as follows and added a couple of references:

The section of the discussion on p. 15 has been renamed to

“A CSR-1 pathways in *Ascaris* in the absence of a piRNA pathway”

The section on p. 16 has been modified to

“Why does AsCSR-1 target so many mRNAs? Does this indicate it still plays a role in “licensing” gene expression in *Ascaris*? In the absence of piRNAs, perhaps the *Ascaris* WAGOs or other genome regulatory processes involved in gene silencing (including programmed DNA elimination [96]) must be counteracted. Alternatively, the AsCSR-1 pathway may not be involved in “licensing gene expression”. Instead, it functions in fine-tuning gene expression [92] or reducing noise to prevent epimutations [97].”

3. The fact that the Ascaris Wagos target repeats is really interesting but needs a little more exploration in the text of the manuscript. In particular, can the authors quote in the manuscript the number of repeat targets compared to the number of protein coding gene targets, for all the IPs (including CSR1) and then carry out a statistical test of the enrichment for either repeats or protein coding genes in each case depending on the number of base pairs of protein coding genes and repeats respectively. I.e. if the genome

is made up of 40% repeats, 10% protein-coding genes, and the wago targets are 99% repeats then this would likely be a significant enrichment using chi squared tests or Fisher's exact test (80%, as total bp that are available is repeats+protein coding genes, vs 99%).

Additionally, it would be helpful if the authors could list some examples of the few genes that are targeted by wago and make some comment about what these are (they may be misannotated repeats).

Finally it would be very useful for the authors to test whether there is still enrichment for repeats when considering only genes and repeats that are within H3K9me2/3 chromatin regions- is the apparent enrichment for repeats just because the elements within these regions are vastly more likely to be repeats?

These analyses are really important. If they hold and the enrichment for repeats is still very significant, then the authors need to point out in the discussion that this is different from *C. elegans*. My suspicion is that the authors will find that the enrichment, particularly when controlling for chromatin regions, is not significant and so the difference to *C. elegans* may be simply due to the fact that the *C. elegans* genome is a lot less repetitive; if this is the case then the authors would have to make that point instead.

The reviewer raises the point whether *Ascaris* WAGOs small RNAs specifically target repetitive sequences or this merely reflects a higher probability that WAGOs target repeats due to the large number of repetitive sequences in *Ascaris* (perhaps compared to *C. elegans*). The most abundant small RNAs are associated with WAGO-1 (see Figure 2 and Figure 4). WAGO-2 small RNAs largely overlap with WAGO-1. NRDE-3 small RNAs also target repeats and the majority of NRDE-3 small RNA targets overlap with WAGO-1/2 targets. We do not think the targeting of these WAGOs to repeats is a consequence of the large number of repeats in the *Ascaris* genome based on the following observations. 1) Our data demonstrate that only 9.5 Mb or ~3% of the genome (compared to 40% of repeats identified in the genome) are high-confidence WAGO targets (see Figure 6, Figure S5 and Table S3), indicating that WAGOs targets a specific subset of repetitive sequences. 2) In the *Ascaris* germline, a 120 bp tandem repeat comprises ~30 Mb or 10% of the genome. This repeat is ~25% of all broadly defined repeats in *Ascaris*. Notably, this repeat and other satellite or short repeats are not targeted by small RNAs. 3) >90% of the defined non-satellite repeats are not targets of WAGO small RNAs (see Figure 6B). Thus, it appears that the WAGOs only target a small select group of the defined repeats. 4) The definition of different types of sequences in the genome and their annotation is a function of the criteria and cutoffs used in the analyses. In our analyses, ~30% of the WAGO targets were not annotated as "traditional" repeats (Figure 6B). This may be due to mis-annotation of the sequences as repeats or that WAGO can target some non-repetitive sequences. Overall, the data indicate that the WAGOs target a specific and small subset of repeats and the existence of large proportion of repeats in the *Ascaris* genome is not a likely explanation for why WAGOs small RNAs target repeats.

Consistent with this, we have carried out the statistical analyses the reviewer suggested. These analyses demonstrate that WAGOs do not randomly target repeats and the targeting is not a function of the genome size or composition (see uploaded Table Reviewer2_Q3_table). Their targets are a specific subset of repeats. We also note that our estimation of the repeat content of *Ascaris* is very liberal (~40%) and could be significantly less. Interestingly, the genomic regions enriched with H3K9me3 also show major overlap with the defined repeats and the WAGO targets (Figure 6B). There is a similar relationship of H3K9me3 enriched regions and the WAGO small RNAs - a large portion (but not all) of the WAGO small RNAs are associated with H3K9me3. Taken together, our data showed that WAGO-1/2 small RNAs are very specific to a small percentage (3%) of the genome that are largely (but not completely) overlap with defined repeats and H3K9me3 regions. Some *C. elegans* WAGO associated small RNAs target repeats, pseudogenes, and aberrant transcripts. However, AsWAGO-1/2 are highly specific for repetitive sequences.

WAGO-1 and WAGO-2 small RNAs also target transcripts. However, most of these transcripts are hypothetical predictions and few of them are clear protein encoding transcripts. The WAGO-1 Ab is highly specific producing very robust IPs with very little noise. Our data suggests that WAGO-2 targets are similar to WAGO-1. Some subtle differences are observed, but these differences could also be due to the quality and affinity of the antibody to these two WAGOs. Future study that is beyond the scope of this work will need to distinguish these differences. Data from WAGO-1 IP suggest that less than 10% of the small RNA

targets are protein encoding transcripts. We believe that a large portion of these transcript targets could be defined as WAGO repeats if the criteria (10x enrichment) for WAGO1/2 targets becomes less stringent. We note a small fraction of the genome could be defined as both repeats and genes. In addition, in any IP experiment, there is a certain degree of noise. This small percentage of targets to protein transcripts could also be due to the IP noise. Given these complications, we were trying to interpret and present the big picture for the majority (80-90%) of the data.

The following has been added to the results text on p. 10 to address the points raised by the reviewer: "The *Ascaris* genome has a larger content (40%) of defined repetitive sequences compared to *C. elegans*. However, the AsWAGO-1/2 targeting of repeats is not a consequence of the increased repeat content as less than 8% of these repeats (3% of the genome) are high confidence AsWAGO-1/2 small RNAs targets."

Minor points

1. "Genome-wide analysis reveals AsCSR-1 small RNAs target a large, comprehensive set of genes" The word *comprehensive* doesn't make sense here- please remove

p. 13, Removed

2. Can the authors replace "complementary" to antisense when they mean small RNAs that are antisense to particular targets? *Complementary* is ambiguous whereas *antisense* I feel is more direct.

p. 8, 9, 15, 23 Replaced

3. In the discussion, the WAGO targets are described as "exclusively" repeats. This needs to be reevaluated in the context of the findings from the reanalyses described in point 3 above, in case they show something slightly different here.

We have changed this from "exclusively" to "largely".

Reviewer #3

Comments/suggestions

Citation needed: "... nuclear Argonautes (HRDE-1 or NRDE-3) are present in many Caenorhabditids and some Clade III nematodes, but they appear absent in other Clades.

This statement is based on our phylogenetic analysis of Argonaute proteins in other nematodes and by others noted in literatures. We have added the following citations:

"and nuclear Argonautes (HRDE-1 or NRDE-3) are present in many *Caenorhabditids* and some Clade III nematodes, but they appear absent in other Clades [8, 106, 109, 110, 112]."

Citation needed: "...Secondary siRNAs (22G-RNAs or longer) are present in Clades III-V, but they are absent in Clades I-II."

We have added the following citations:

"Secondary siRNAs (22G-RNAs or longer) are present in Clades III-V, but they are absent in Clades I-II [8, 89, 90, 111, 112]."

It would be worth a table/mention of the % identity of each Ascaris Argonaute to the C.elegans orthologs they are named after.

We have chosen not to include such a table. % identity is often not as informative as phylogenetic analyses and can be misleading, particular when dealing with key functional regions vs. overall conservations. In addition, BLAST analyses and other features of the proteins that are informative have been used in our analyses.

Figures

Figure 1 - From the phylogeny, why is CO4F12.1 called the CSR-1 clade? And how do the authors claim "AsWAGO-3 appears orthologous to C. elegans C04F12.1 based on the phylogenetic analysis"

AsWAGO-3 in several phylogenetic analyses clusters with *C. elegans* CO4F12.1, CSR-1, and AsCSR-1. In addition, BLAST analyses indicated it is most closely related to *C. elegans* CO4F12.1 and CSR-1 compared to the other Argonautes. Our small RNA IP and sequencing data also indicates that AsCSR-1 and AsWAGO-3 bind to largely overlapping small RNAs targeting mRNAs. We have altered the text to state

“AsWAGO-3 is most likely orthologous to *C. elegans* C04F12.1.”

Figure 7 – A: the claim from this figure is that “AsNRDE-3 bound small RNAs change their targets from genomic repetitive sequences to mRNAs during spermatogenesis. However showing the first nt plots this way does not really show quantitative changes. I think an additional plot, like one shown in figure 6 with the proportion of each class of RNA/genome region, would help.

We agree with the reviewer. We have added a plot that shows the changes for the quantitative number of mRNAs and repeats targets for AsNRDE-3 during spermatogenesis. The plot (new Fig. 7B, shown here) illustrates the decrease in repetitive sequences targeted by AsNRDE-3 and the dramatic increase in mRNAs targets in the late meiosis (M5-M7).

Legend: B. AsNRDE-3 mRNA and repeat target changes by during spermatogenesis.

Figure 7 C- The schematic is a hard to follow (also not clear what the arrows mean in relation to the inhibition signal?).

We have updated the legend as follow which is now D: “D. Model for *Ascaris* small RNA pathways (see text). The two concentric circles represent *Ascaris* mRNAs (left) and repeats (right) targeted by small RNAs. The larger size of the mRNA circle indicates the larger number, complexity, and abundance of the mRNAs compared to the repeats targeted by small RNAs. Overall, however, in terms of abundance of small RNAs, 80-90% of all small RNAs target repeats. The 10 *Ascaris* Argonautes and their known associated major small RNAs (22G, 23G, 24G, 26G, and miRNAs) are shown. Lines with an arrow indicated licensing or fine tuning (→) of expression, a blocking line indicates repression (—|), and an arrow with a blocking line indicates both licensing/tuning and repression (→|). Bolder lines indicate predicted stronger affects. Note several *Ascaris* Argonautes (CSR-1, WAGO-3 and NRDE-3) could be involved in both licensing and repression of their mRNA targets. In addition, NRDE-3 and WAGO-3 can target both mRNAs and repeats and their targets change through development, illustrating the plasticity of small RNA pathway in *Ascaris*.”

Supplementary figures

In general a bit more information in the figure legends or annotation/labelling in the axes of graphs is merited.

We have updated the figures and legends with more information and descriptions shown below.

Sfig 1: it might be worth better defining/annotating CSR-1 on this figure – what are all of the gene models?

We have updated the figure with an arrow pointing to the CSR-1 gene. The legend has been updated as follows:

“A genome browser view on Chr17 shows the AsCSR-1 gene (in between 2.9 - 2.95 Mb, marked by an arrow), other nearby genes, RNA-seq and H3K9me3 ChIP-seq from selected developmental stages. The gene models are largely based on RNA-seq (including ISO-seq) data. Red and blue indicate forward and reverse transcription, respectively. Note the high expression of AsCSR-1 in early embryos (1-4 cells) and the loss of its expression in 32-64 cells associated with heavy H3K9me3 marks.”

Sfig 2: From these data - I do not draw the same conclusions on nuclear/cytoplasmic relations as the authors. From these blots, WAGO-3 is not shown and WAGO-2 seems only to be found in nucleus. But the authors state “AsWAGO-3, AsCSR-1, and AsNRDE-3 are present in both the cytoplasm and nuclei, but significantly greater in the cytoplasm” “WAGO-2 is equally present in the cytoplasm and nucleus whereas AsWAGO-1 is around two-fold higher in the cytoplasm compared to the nucleus.”?

The reviewer is correct. We have modified the text on page 8 as follows:

“AsCSR-1 and AsNRDE-3 are present in both the cytoplasm and nuclei, but significantly greater in the cytoplasm. Notably, AsALG-1 is also present in nuclei and is associated with miRNAs. However, AsWAGO-2 is predominantly in the nucleus whereas AsWAGO-1 is slightly higher in the cytoplasm compared to the nucleus.”

Sfig 4: can the authors comment on why more ribosomal in testis region 4?

We suggest this may be due to the degradation of rRNAs in the late meiosis. We have added the following to the legend:

“We observed an increased amount of small RNAs matching rRNAs in region 3 that have a random size distribution, consistent with what is observed in spermatids (Fig. 4C). This may be associated with the turnover of rRNAs in *Ascaris* late meiosis.”

Page 5 of 7 Sfig 8: More legend would help - for example what “RNA” in (A) on Y axis is referring to and how the data are normalized.

For Fig. S5, we have added the following in the legend:

“Distribution of small RNAs associated with A) *Ascaris* CSR-1 (green) and B) WAGO-1 (Red) Argonautes on *Ascaris* chromosomes. The small RNAs are normalized to the total number of sequencing reads. The scale on Y-axis is capped so the plots at highly enriched regions will not extend beyond the top lines. “

Fig. S6, has been revised as follows

“Distribution and strandedness of small RNAs to mRNAs. Meta-analysis of siRNA distribution across mRNAs (in 100 bins on x-axis). The relative levels of the siRNAs are shown on y-axis. Note the biased antisense targeting at the 5'-end of mRNAs at late pachytene and meiosis for many of the Argonaute associated small RNAs, suggesting a possible concerted effort to repress the mRNAs for their clearance. Blue represents antisense to mRNAs and red indicates sense mRNAs.”

For Fig. S8, we have added the following in the legend:

“A. Repetitive sequences targeted by small RNAs in the testis (data from Figure 2A). The small RNAs are normalized to the total number of sequencing reads (see methods). The RNA level (in rpkm) for these

repeat targets on the y-axis is derived from testis RNA-seq data. Overall, the RNA expression level from these repeats is low, suggesting a potential repressive role of the siRNAs.”

Sfig 9: It may be worth clarifying if the Y axis (after the top, mRNA) that simply say the AGO protein name are in fact small RNA reads from IP datasets – in general it should not be assumed that readers can jump in and follow at the level of the authors.

We have revised and added the following in the legend:

“Genome browser tracks showing four Argonaute mRNAs targeted by their siRNAs in different developmental stages. The mRNA levels are normalized RNA-seq data from the M1-M8, ovary, and 4-cell embryos. PRO-seq data shows the nascent RNAs for these four Argonaute genes. The data below the PRO-seq are small RNA sequencing data from the corresponding Argonaute IPs. This illustrates potential auto-regulation of the expression of Argonautes by their associated small RNAs.”

Methods

“We normalized the coverage for each library to 30 million reads, a number close to the average number of raw input reads for the libraries”. Slightly unclear: does this mean they subsampled the data or how exactly is this justified?

Each library is converted to a total number of 30 million reads. For example, if the sequencing depth for a library is 10 million, the number is multiplied by 3 and if a library is 60 million, it is multiplied by 0.5. This is not a subsampling. The goal was to normalize the libraries to analyze the same number of reads for comparisons.

Additional

Some of the conclusions about WAGO expression and small RNA expression would be supported by a bit more detail: in the context of potentially different starting material amounts or affinity of antibodies, is there a quantitative way to claim that AsWAGO-1 and its small RNAs are expressed at much higher levels than AsWAGO-2?

WAGO-1 and WAGO-2 mRNA expression is similar. However, we do not know the levels of these WAGO proteins. Equal amounts of starting material were used wherever possible as well as the same amount of affinity purified antibody. However, the affinity and quality of independently made antibodies differ. For example, WAGO-1 is a very strong antibody with what appears to be high affinity. Our data suggests that WAGO-2 targets are very similar to WAGO-1. However, the amount of small RNAs from a WAGO-2 IP are significantly lower compared to WAGO-1. Because all antibodies are unique, we try not to make quantitative comparisons between antibodies. The data presented and compared are from mRNA-seq and small RNA-seq.

We have adjusted the text on p. 9 as follows to reflect the reviewer’s comment and our data

“WAGO-1 and WAGO-2 are expressed at similar levels and their associated small RNA targets are highly similar.”

Sample IDs	# of Targets		Chi Square & p value	
	genes	repeats	χ^2	p
ALG1_64hr	9	54	18.01	<0.001
ALG1_ovary	11	98	41.97	<0.001
ALG1_male	35	119	20.29	<0.001
ALG4_64hr	39	523	266.71	<0.001
ALG4_ovary	1193	937	224.59	<0.001
ALG4_M1	734	1040	0.57	0.4485
ALG4_M2	1426	1016	345.79	<0.001
ALG4_M3	1128	949	173.33	<0.001
ALG4_M4	1360	1001	304.68	<0.001
ALG4_M5	1070	915	155.68	<0.001
ALG4_M6	1206	363	896.58	<0.001
ALG4_M7	1009	374	626.42	<0.001
CSR1_64hr	3864	377	5058.34	<0.001
CSR1_ovary	3770	594	4294.17	<0.001
CSR1_M1	827	1066	8.32	0.0039
CSR1_M2	2882	963	2100.85	<0.001
CSR1_M3	3168	826	2782.24	<0.001
CSR1_M4	3133	849	2682.22	<0.001
CSR1_M5	2166	778	1444.88	<0.001
CSR1_M6	2076	607	1625.29	<0.001
CSR1_M7	1871	552	1444.85	<0.001
INPUT_64hr	173	681	148.65	<0.001
INPUT_ovary	245	831	144.57	<0.001
INPUT_M1	573	2000	380.55	<0.001
INPUT_M2	434	1504	277.87	<0.001
INPUT_M3	332	1352	315.96	<0.001
INPUT_M4	309	1316	325.23	<0.001
INPUT_M5	356	1290	254.36	<0.001
INPUT_M6	1329	1196	164.65	<0.001
INPUT_M7	826	913	36.87	<0.001
NRDE3_64hr	318	828	79.83	<0.001
NRDE3_ovary	617	843	1.92	0.1655
NRDE3_M1	423	1365	220.95	<0.001
NRDE3_M2	405	1234	177.32	<0.001
NRDE3_M3	216	886	205.81	<0.001
NRDE3_M4	79	759	343.42	<0.001
NRDE3_M5	1232	1144	135.04	<0.001
NRDE3_M6	2050	841	1197.68	<0.001
NRDE3_M7	1573	752	756.1	<0.001
WAGO1_64hr	134	716	220.85	<0.001
WAGO1_ovary	170	913	284.68	<0.001

WAGO1_M1	496	2179	575.47	<0.001
WAGO1_M2	313	1674	532.75	<0.001
WAGO1_M3	236	1399	481.69	<0.001
WAGO1_M4	170	1155	436.22	<0.001
WAGO1_M5	183	1277	492.77	<0.001
WAGO1_M6	219	1322	460.61	<0.001
WAGO1_M7	196	1266	463.19	<0.001
WAGO2_64hr	60	574	257.82	<0.001
WAGO2_ovary	170	767	199.37	<0.001
WAGO2_M1	273	1371	407.13	<0.001
WAGO2_M2	147	964	353.22	<0.001
WAGO2_M3	109	892	374.15	<0.001
WAGO2_M4	128	941	371.51	<0.001
WAGO2_M5	60	732	363.94	<0.001
WAGO2_M6	77	392	114.36	<0.001
WAGO2_M7	44	297	108.81	<0.001
WAGO3_64hr	3406	430	4117.04	<0.001
WAGO3_ovary	909	753	144.89	<0.001
WAGO3_M1	234	1083	292.64	<0.001
WAGO3_M2	183	928	274.3	<0.001
WAGO3_M3	448	983	52.25	<0.001
WAGO3_M4	192	886	237.12	<0.001
WAGO3_M5	379	847	48.4	<0.001
WAGO3_M6	288	478	2.75	0.097
WAGO3_M7	46	288	99.96	<0.001

$df=1$

H_0 : There is no difference in targeting genes vs repeats by smRNAs from IP.

The null hypothesis was rejected with confidence level of p-value 0.05 and $df=1$ in all cases except
The results of tests indicate non random targeting of genes and repeats for corresponding IPs in

NOTE: in Targets 121nt tandem repeat was counted as 1. The nucleotide variation in 121 bp is li

NOTE: see Chi² test calculated for each IP sample calculated for Number of Targets; Number of

Total number of gene : 15714

Total number of repeats: 23073

Nucleotides in bp		Chi Square & p value (no 121 bp)		Chi Square & p va
genes	repeats	χ^2	p	χ^2
175106	235651	233213.77	<0.001	115144.16
126888	115915	68277.73	<0.001	25447.92
431493	91774	14965.7	<0.001	53957.71
770565	1859979	2994650.05	<0.001	1757413.36
21927853	3471325	1880583.78	<0.001	4528717.19
14145281	3089833	472041.21	<0.001	1789260
26223522	2922410	3854703.57	<0.001	7496042.49
21081355	2399778	2953012.55	<0.001	5810585.17
24339144	2347733	4096393.76	<0.001	7561648.13
18213495	1484256	3416406.89	<0.001	6030406.87
16263543	394099	4812685.35	<0.001	7340554.72
13264017	425462	3653936.68	<0.001	5678498.3
68598880	355063	29678611.72	<0.001	42266928.53
67217860	555272	28271475.24	<0.001	40542882.59
14172219	2178104	1239160.7	<0.001	2923419.61
51222470	2375246	15038364.24	<0.001	23682501.88
55596575	1488658	19332130.65	<0.001	29024232.84
54518677	1403905	18996542.38	<0.001	28476856.77
37486074	851696	12309230.03	<0.001	18516966.13
33068112	907817	10277715.87	<0.001	15657045.38
29622630	529895	9725859.28	<0.001	14542146.36
3149758	3615611	3043847.57	<0.001	1373319.66
4375826	3678919	1932530.08	<0.001	647188.69
10316226	4873261	453364.74	<0.001	1761.12
8146836	4733119	1029239.48	<0.001	106867.66
6486151	4732683	1889319.74	<0.001	490688.18
6067827	4471246	1819407.44	<0.001	483120.25
5939988	4154710	1497793.72	<0.001	350727.65
20279503	3318009	1613871.86	<0.001	4006919.52
11503565	2690689	278179.23	<0.001	1259895.7
5980753	3487140	756810.18	<0.001	80734.23
10492346	3571035	2443.08	<0.001	347019.82
7147672	4531713	1276644.3	<0.001	214768.84
7036109	4524291	1324400.37	<0.001	235981.61
3602639	3400772	2168807.54	<0.001	834760.08
1485923	2183562	2395890.87	<0.001	1227235.42
20233930	3256850	1664931.94	<0.001	4072281.3
32116000	1814634	7933721.65	<0.001	12960404.38
24504021	1398525	5819536.59	<0.001	9558859.08
2197352	3821999	4931668.83	<0.001	2675541.38
3054037	3988774	3919883.46	<0.001	1902475.85

8278323	5633948	1929667.12	<0.001	417433.44
5330014	5407741	3907358.43	<0.001	1605992.65
4331953	5088701	4456625.69	<0.001	2037881.5
3205718	4649237	5118304.06	<0.001	2603348.44
3370432	4411192	4357700.27	<0.001	2116501.91
4192023	4325062	3173782.87	<0.001	1324116.28
3779646	4125648	3266202.9	<0.001	1423279.45
856813	3302902	6740846.09	<0.001	4213183.18
2732730	3741123	3860413.74	<0.001	1916424.85
5087386	3070014	738517.09	<0.001	97842.72
2554474	2783191	2175909.44	<0.001	947920.88
1938428	2602111	2609502.36	<0.001	1284733.39
2146789	3276981	3755828.51	<0.001	1950898.74
1040016	2292602	3493329.44	<0.001	2013431.06
1077578	725385	230462.58	<0.001	48447.26
668730	716955	540019.13	<0.001	232852.87
61354938	562904	24883970.94	<0.001	35891518.9
18221812	1963814	2675008.84	<0.001	5153665.5
4262430	2256700	342407.95	<0.001	12661.66
3255540	2314382	851502.83	<0.001	208348.18
8290231	2236202	75919.56	<0.001	638367.71
3620376	2427879	776686.02	<0.001	161037.49
6760443	1546722	177763.64	<0.001	758204.99
3532025	786670	103175.58	<0.001	411510.1
763280	648722	338715.31	<0.001	115462.35

pt for ALG4_maleR1a, NRDE3_female, WAGO3_maleR3c for "# of targets".

both cases - for number of targets and for number of targeted nucleotide bps (with or witho

mitted. In addition, no signal was observed for any smRNAs IPs from 121bp repeat.

f nucleotides (with/without 121 bp repeat)

<0.001
<0.001
<0.001
<0.001
<0.001
<0.001
<0.001
<0.001
<0.001
<0.001
<0.001
<0.001
<0.001
<0.001
<0.001
<0.001
<0.001
<0.001
<0.001
<0.001
<0.001
<0.001
<0.001
<0.001
<0.001
<0.001
<0.001
<0.001
<0.001
<0.001

out 121bp tandem repeat).

REVIEWERS' COMMENTS

Reviewer #1 (Remarks to the Author):

My comments were adequately addressed in the revised manuscript.

Reviewer #2 (Remarks to the Author):

Thank you for responding comprehensively to my questions from the first round. I look forward to seeing the manuscript published.

Reviewer #3 (Remarks to the Author):

The authors have addressed all my comments.